# Laboratory mice engrafted with natural gut microbiota possess a wildling-like phenotype

Solveig Runge [1,2,3,11] ✉, Silvia von Zedtwitz [3,11], Alexander M. Maucher [3,4], Philipp Bruno[1,2], Lisa Osbelt [5], Bei Zhao[3], Anne M. Gernand [3], Till R. Lesker[5], Katja Gräwe[6], Manuel Rogg [6], Christoph Schell[6], Melanie Boerries [7,8], Till Strowig [5,9,12] ✉, Geoffroy Andrieux [7,12] ✉, Benedikt Hild [10,12] ✉ & Stephan P. Rosshart [1,2,3,12] ✉

Conventional laboratory mice housed under specific pathogen-free (SPF) conditions are the standard model in biomedical research. However, in recent years, many rodent-based studies have been deemed irreproducible, raising questions about the suitability of mice as model organisms. Emerging evidence indicates that variability in SPF microbiota plays a significant role in data inconsistencies across laboratories. Although efforts have been made to standardize microbiota, existing microbial consortia lack the complexity and resilience necessary to replicate interactions in free-living mammals. We present a robust, feasible and standardizable approach for transplanting natural gut microbiota from wildlings into laboratory mice. Following engraftment, these TXwildlings adopt a structural and functional wildling-like microbiota and host physiology toward a more mature immune system, with characteristics similar to those of adult humans. We anticipate that adopting wild mouse-derived microbiota as standard for laboratory mouse models will improve the reproducibility and generalizability of basic and preclinical biomedical research.

Conventional specific pathogen-free (SPF) laboratory mice (lab mice) are the standard model in global biomedical research and have been prerequisites for Nobel Prize discoveries such as MHC restriction[1,2] and immune checkpoint inhibition[3]. Despite these achievements, emerging evidence indicates that many published results are valid only under narrow conditions and cannot be reproduced by other research institutions[4–9]. In the USA alone, over half of the preclinical research studies involving animal models are considered irreproducible, representing a cost of $28 billion USD per year[10]. A substantial body of literature demonstrates that divergent microbiota among commercial

[1]Department of Microbiome Research, University Hospital Erlangen, Friedrich-Alexander-Universität Erlangen-Nürnberg (FAU), Erlangen, Germany. [2]Department of Medicine 1, University Hospital Erlangen, Friedrich-Alexander-Universität Erlangen-Nürnberg (FAU), Erlangen, Germany. [3]Department of Medicine II, Medical Center - University of Freiburg, Faculty of Medicine, Freiburg, Germany. [4]Friedrich-Alexander-Universität Erlangen-Nürnberg (FAU), Erlangen, Germany. [5]Department of Microbial Immune Regulation, Helmholtz Center for Infection Research, Braunschweig, Germany. [6]Institute of Surgical Pathology, Medical Center - University of Freiburg, Faculty of Medicine, Freiburg, Germany. [7]Institute of Medical Bioinformatics and Systems Medicine, Medical Center - University of Freiburg, Faculty of Medicine, University of Freiburg, Freiburg, Germany. [8]German Cancer Consortium (DKTK), Partner site Freiburg, a partnership between DKFZ and Medical Center – University of Freiburg, Freiburg, Germany. [9]Center for Individualized Infection Medicine (CiiM), a joint venture between the Helmholtz-Center for Infection Research (HZI) and the Hannover Medical School (MHH), Hannover, Germany. [10]Department of Gastroenterology, Hepatology and Transplant Medicine, Medical Faculty, University of Duisburg-Essen, Essen, Germany. [11]These authors contributed equally: Solveig Runge, Silvia von Zedtwitz. [12]These authors jointly supervised this work: Till Strowig, Geoffroy Andrieux, Benedikt Hild, Stephan P. Rosshart. ✉e-mail: Solveig.Runge@uk-erlangen.de; Till.Strowig@helmholtz-hzi.de; Geoffroy.Andrieux@uniklinik-freiburg.de; Benedikt.Hild@uk-essen.de; Stephan.Rosshart@uk-erlangen.de

vendors and research institutions is a primary reason for this global reproducibility crisis[11–25]. It is crucial to recognize that the combination of the microbiome and the host genome drives the mammalian phenotype[17]. Consequently, divergent microbiota lead to divergent immune phenotypes and ultimately, to irreproducible and contradictory data across different institutions[15,21,26–29]. This poses a significant burden to scientific progress.

This fundamental problem can be attributed to the origin of conventional laboratory mice. Their microbiota are the result of repeated germ-free rederivation and recolonization in restrictive laboratory environments leading to a complete loss of naturally co-evolved microbes. Consequently, lab mice microbiota lack the complexity, evolutionary adaptation, and resilience of natural microbiota. This is evident in the divergent composition of their microbiome[30–39] and the underrepresentation of key non-bacterial constituents such as the virome and mycobiome[31,33,40]. In addition, they are rapidly outcompeted and replaced by natural microbiota in a co-housing setup[31,34] and undergo significant changes in community structure upon even minor environmental perturbations, such as mouse husbandry conditions[12] and transportation[14,22], resulting in divergent and location-specific conventional lab microbiota and phenotypes across different institutions[41]. The current strategies to address the reproducibility crisis focus on documentation and standardization[18,42,43]. However, documentation of the microbiota composition and all influencing factors does not correct the underlying problem of divergent microbiota. Further, the multitude of factors influencing the microbiota – such as temperature[44,45], diet, bedding materials[12,46], water pH and treatment[21] – make it nearly impossible to account for every variable[17,18]. Hence, a retrospective analysis will rarely clarify the differences observed[19,47–49]. Therefore, conventional SPF microbiota, cannot be used for standardization since they lack the required complexity and resilience, resulting in the re-emergence of divergent microbiota and irreproducible data across different institutions. Further, defined microbial consortia can solve the problem of divergent microbiota across different institutions[47–49]. However, these artificial consortia remain stable only under gnotobiotic housing conditions and lack the complexity of a naturally co-evolved microbiota. As a result, these approaches prioritize reproducibility at the expense of physiological relevance. Therefore, the goal must be to select a standard with the necessary complexity and resilience to overcome narrow, non-generalizable local phenotypes and create robust mouse models across different institutions[41].

We suggest that naturally co-evolved microbiota of wild mice possess the necessary biological complexity alongside unique characteristics to serve as a suitable candidate for successful standardization[41]: I) They can be harvested, viably preserved, frozen, bio-banked, thawed, transferred, and engrafted into lab mice[30]. II) Natural microbiota are well characterized in their overall composition[31] and have evolved under evolutionary pressure in a challenging environment. As a result, they are highly adapted to the mouse gut, III) remain stable in the multi-generational offspring of lab mice[30,31], IV) outcompete lab microbiota and possess remarkable resilience against strong environmental disturbances like microbial challenges[31], antibiotics[31], change of diet[31,34,35], and a stable core microbiome upon shipment and across multiple institutions with different husbandry conditions[30,31,33,35,50]. V) Finally, natural microbiota-based mouse models, such as wildlings, have a superior translational research value over lab mice since their mature immune system better mirrors human physiology[31,50–53]. However, creating natural microbiota-based models demands significant funding, expertise, and specialized infrastructure, making them inaccessible for widespread use[30,31,44,51,52,54].

In this work, we develop and validate a standardizable, feasible, and controlled natural gut microbiota transplantation system (TX system) to transform conventional lab mice into natural microbiota-based models through a single oral gavage with natural gut microbiota sourced from wildlings. Mice generated through this system (TXwildlings) may be easily adopted in standard mouse facilities with minimal technical requirements, enabling the rapid and widespread use of natural microbiota-based models. This approach may help to resolve the reproducibility crisis and enhance the translational success of basic and preclinical biomedical research.

## Results

### Natural gut microbiota of wildlings outcompete lab microbiota

The creation of natural microbiota-based mouse models demands significant funding, expertise, and specialized infrastructure like BSL3 laboratories, outdoor enclosures or specialized indoor enclosures[52,54,55]. To circumvent this, our study aimed to develop and validate a feasible TX system to convert lab mice into natural microbiota-based models. To test this, we harvested the natural gut microbiota of wildlings and performed one oral gavage on one out of five adult, fully colonized C57BL/6 mice from Taconic Biosciences, housed in a single cage. Fecal pellets of all mice were collected, starting from day 0 until day 28 and 16S rRNA gene profiling was performed to assess changes in microbiota composition over time. The lab mouse group exposed to wildling gut microbiota was named TXwildlings and compared to their lab counterparts as well as wildlings (Fig. 1a upper panel). We found that the natural gut microbiota of wildlings swiftly outcompeted the gut microbiota of all five fully colonized adult lab mice (Fig. 1b). More importantly, by day 28 post-transplantation the gut microbiota composition of all TXwildlings resembled that of wildlings and differed significantly from Taconic lab mice (Fig. 1b, c, Supplementary Fig. S1a, e). Importantly, the microbiota of both, mice receiving transplantation and those co-housed, converted in a similar fashion (Supplementary Fig. S1c, d). To probe the stability and resilience of natural gut microbiota, we performed the identical experiment in the opposite fashion and engrafted gut microbiota from Taconic lab mice into wildlings, referred to as TXlab mice (Fig. 1A lower panel). Exposure to Taconic lab microbiota had no impact on the composition of the natural microbiota of wildlings (Fig. 1d, e, Supplementary Fig. S1b).

There is an increasing appreciation that non-bacterial constituents of the microbiome such as fungi and viruses play an important physiological role[56–61]. Therefore, we assessed whether non-bacterial components were also successfully transferred to TXwildlings. Using shotgun and internal transcribed spacer (ITS) sequencing, TXwildlings as well as wildlings were found to carry significantly more non-bacterial microorganisms, like protozoan, protists, fungi, and viruses than conventional lab mice (Supplementary Fig. S1f and Supplementary Table 1). In conventional lab mice the fungal biomass was very low, while TXwildlings and wildlings displayed a substantially higher fungal biomass dominated by *Kazachstania pintolopesii* (Supplementary Fig. S1g). Further, pathogenic experience is a wildling-defining characteristic and crucial for the functionality of several natural microbiota-based mouse models[31,51,52]. Hence, we characterized the pathogen profile using serology and PCR and found that we were successful in transferring pathogens from wildlings to TXwildlings (Supplementary Table 2). Of note, TXwildlings as well as the wildling colony established and used by us do not harbor human pathogens and meet biosafety level 1 standards. This suggests a complete engraftment of not only bacteria, but also non-bacterial constituents of the microbiome alongside pathogens.

Next, we assessed whether this phenomenon was specific to Taconic lab mice or could be generalized to cases where natural microbiota compete with other lab microbiota. Hence, we repeated the experiments with mice from the major globally leading vendors (Taconic Biosciences, The Jackson Laboratory, Charles River, Janvier, Envigo). Despite using just a single oral gavage and a numerical disadvantage in a 1:4 co-housing setup, by day 28 post-transplantation the

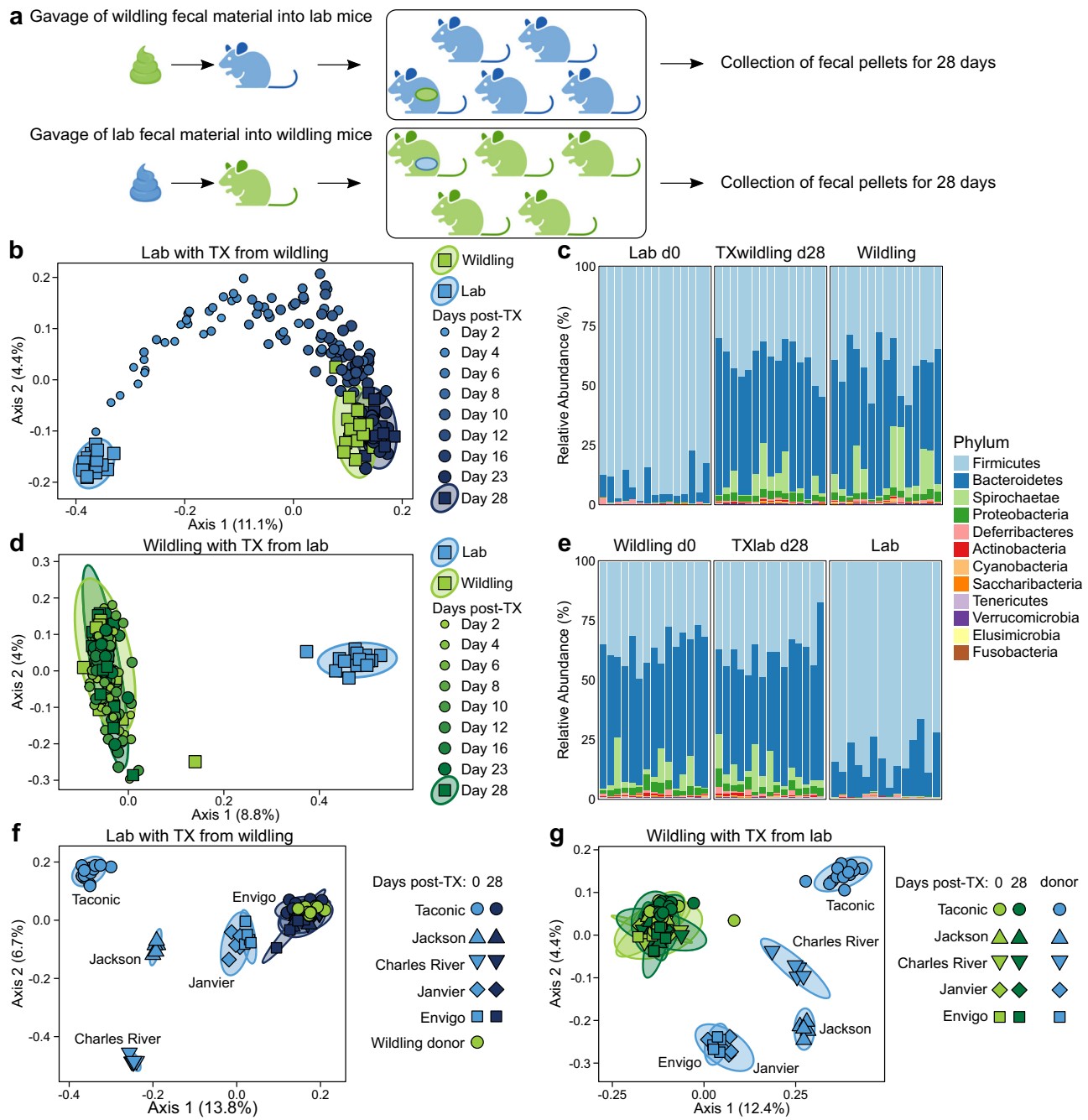

**Fig. 1 | Natural gut microbiota of wildlings outcompete lab microbiota of the major globally leading vendors.** Fecal material from wildlings was transplanted by oral gavage into a lab mouse, which was housed with four lab mice in the same cage. The same experiment was performed with a wildling mouse transplanted with fecal material from lab mice. Fecal pellets from all mice were collected over 28 days and 16S rRNA gene profiling was performed. **a** Schematic representation of the experimental design. **b** PCoA based on beta diversity using Jaccard distance compares the microbiota of wildlings before and at the indicated time points post-transplantation with those of lab donor mice. **c** Relative abundance of the bacterial phyla of lab mice before and 28 days post-transplantation in comparison to the wildling donor mice. **d** PCoA based on beta diversity using Jaccard distance compares the microbiota of wildlings before and at the indicated time points post-transplantation with those of lab donor mice. **e** Relative abundance of the bacterial

phyla of wildlings before and 28 days post-transplantation compared to the lab donor mice. **f** Lab mice were purchased from different vendors and transplanted with wildling fecal material. PCoA based on Jaccard distance shows samples from lab mice of the respective vendors before and 28 days post-transplantation, compared to the samples from the wildling donor mice. **g** Lab mice were purchased from different vendors and used for transplantation to wildlings. PCoA based on Jaccard distance shows samples from wildlings before and 28 days post-transplantation, compared to the samples from the respective lab donor mice. **b–e** Data are from three independent experiments, with five donor mice per transplantation and one recipient mouse co-housed together with four cage mates in each experiment. **f**, **g** Data are from one experiment. Figure in (**a**) was created in BioRender. Bruno, P. (2025) https://BioRender.com/60kbw9g. TX: Transplantation.

gut microbiota composition of TXwildlings from all vendors resembled that of wildlings and differed significantly from all lab mice (Fig. 1f and Supplementary Fig. S2a). This pattern was true for beta diversity, alpha diversity, and phylogenetic composition at the family level

(Supplementary Fig. S2a, c-d). While the baseline microbiota differed significantly between vendors, transplantation increased cross-vendor similarity (Supplementary Fig. S2b–d). Conversely, the exposure to lab microbiota, regardless of vendor origin, did not affect the composition

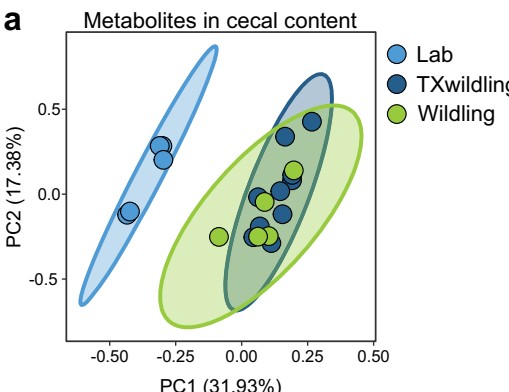

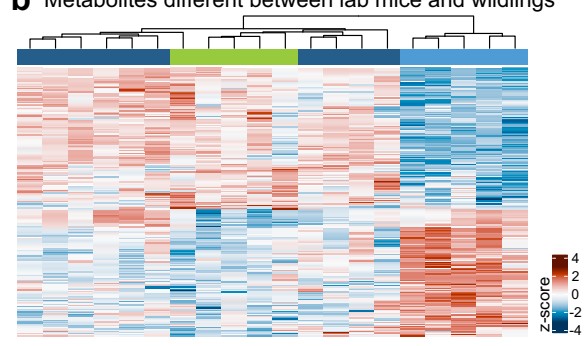

**Fig. 2 | The microbiota of TXwildlings closely resemble the functional capability of wildling microbiota and differ significantly from lab microbiota.** Global metabolomics analysis was performed on the cecal content of lab mice, TXwildlings and wildlings. **a** PCA of all detected metabolites was used to compare lab mice with TXwildlings and wildlings. **b** The heatmap shows the metabolites significantly different between lab mice and wildlings. Color code depicts the row-wise scaled (z-score) metabolite intensity. PC: Principal component.

of the natural microbiota of wildlings (Fig. 1g and Supplementary Fig. S2e).

Taken together, the natural gut microbiota of wildlings outcompete the lab microbiota of the major globally leading vendors. This highlights the superior co-evolutionary adaptation of natural microbiota to the gut niche and supports this as a generalizable phenomenon. Hence, natural gut microbiota can be harvested from wildlings and used in a feasible, controlled, and standardizable transplantation system to convert adult lab mice into TXwildlings carrying natural microbiota.

## TXwildling microbiota mirror wildling microbiota functionality

The 16S rRNA gene profiling demonstrated that the natural gut microbiota of wildlings can swiftly outcompete and replace the microbiota of fully colonized adult lab mice. However, 16S rRNA gene profiling does not provide any functional information. Consequently, the question remained whether the microbiota in TXwildlings merely replicated the community structure or also possessed the functional characteristics of natural microbiota. To answer this question, we conducted a global metabolomics analysis of cecal contents from TXwildlings, wildlings, and lab mice kept under identical husbandry conditions. As illustrated by Principal Component Analysis (PCA), the global gut metabolomics profile of TXwildlings resembled that of wildlings and differed significantly from lab mice (Fig. 2a, Supplementary Fig. S3a and Supplementary Data 1). These findings were further confirmed when focusing only on metabolites that were significantly different between wildlings and lab mice (Fig. 2b). Conversely, exposure to lab microbiota, had no effect on the global gut metabolomics profile of wildlings (Supplementary Fig. S3b–d).

In summary, we found that the microbiota of TXwildlings not only mimicked the community structure of wildling microbiota but also closely resembled the functional capability of natural microbiota.

## Gut features of TXwildlings phenocopy wildlings

TXwildlings are colonized with a microbiome that mimics the community structure and functional capability of natural microbiota. However, wildlings and laboratory mice differ significantly in various aspects of host physiology, especially features of the immune system[31]. Hence, we investigated if TXwildlings would adopt these defining characteristics of wildlings, by conducting a multi-organ systems analysis examining immunological barrier sites (gut and lung), central non-lymphoid organs (liver), lymphoid organs (mesenteric lymph nodes (mLN)), and blood. We started with the assessment of the gut, as it is the immunological barrier site local to

the transplanted natural microbiota. TXwildlings and wildlings had an identical cecum size significantly smaller than the cecum of lab mice (Fig. 3a), while the cecum of TXlab mice remained unaffected (Supplementary Fig. S4a). The colon transcriptome of TXwildlings closely resembled that of wildlings and differed significantly from that of lab mice (Fig. 3b and Supplementary Fig. S4c). This is further illustrated by a comparison of gene expression between TXwildlings and lab mice (Fig. 3c left panel), as well as between TXwildlings and wildlings (Fig. 3c right). While there were 1422 genes significantly differently regulated between TXwildlings and lab mice, only two genes significantly differed between TXwildlings and wildlings. This indicates that the colon transcriptome of TXwildlings is highly similar to that of wildlings. In contrast, exposure of wildlings to lab microbiota did not result in any discernible impact on the colon transcriptome (Supplementary Fig. S4b, d, e). A Gene Set Enrichment Analysis (GSEA) revealed an enrichment of immune-related gene sets, including those associated with the response to symbionts, viruses, and the type I interferon response in TXwildlings (Fig. 3d and Supplementary Fig. 4f). Lab mice exhibited an enrichment of tissue homeostasis pathways, including translation- and mitochondria-related pathways (Fig. 3d and Supplementary Fig. 4f). To gain further insight into the immune landscape of the gut, concentrations of key cytokines and chemokines were measured in the colon. TXwildlings exhibited higher cytokine and chemokine levels than lab mice, matching those of wildlings. Interestingly, IL-33 was initially elevated in lab mice and reduced after the transplant to wildling levels. Together, the overall cytokine and chemokine profile of TXwildlings resembled that of wildlings and differed significantly from lab mice (Fig. 3e and Supplementary Data 2). The opposite experiment did not affect the cytokine and chemokine profile of wildlings (Supplementary Fig. S4g).

In conclusion, our TXsystem not only creates TXwildlings with a functional and structural natural microbiome but also induces a wildling-like phenotype in the local gut barrier site, including anatomical features, global transcriptomes, and cytokine and chemokine profiles.

## TXwildlings adopt systemic traits characteristic for wildlings

Next, we assessed whether the TX system had the capability of inducing wildling-defining characteristics systemically. Therefore, we analyzed central non-lymphoid organs, lymphoid organs, and distant immunological barrier sites. Considering the high rate of microbial antigen transport through abdominal lymphatic tissues and the significant blood flow from the intestine to the liver, we proceeded analyzing the mLN and liver.

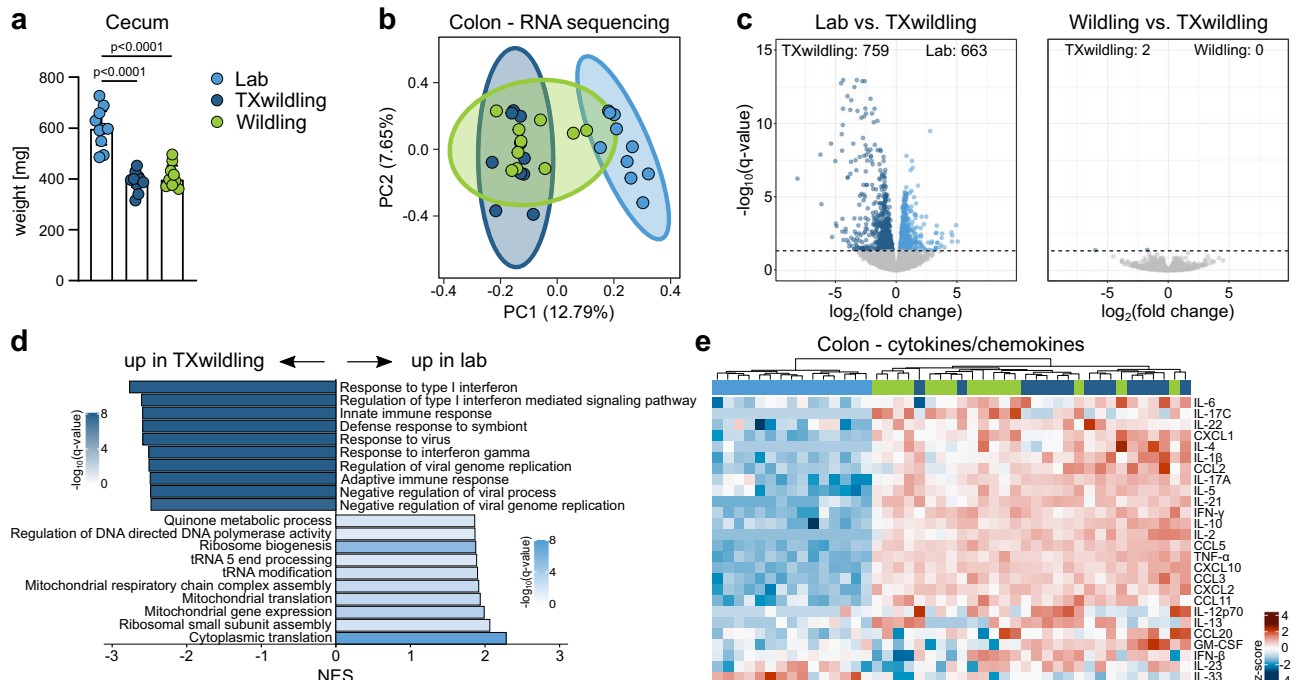

**Fig. 3 | The gastrointestinal tract of TXwildlings phenocopies wildlings and differs significantly from lab mice.** The engraftment of fecal material from wildlings into lab mice resulted in the TXwildlings group, which was then compared to lab mice and wildlings. **a** Weight of the cecum in lab mice, TXwildlings, and wildlings. **b–d** RNA sequencing was performed on colon tissue from lab mice, TXwildlings, and wildlings. **b** PCA of samples from the three groups. The ellipses represent 95% confidence intervals. **c** Volcano plots compare lab mice and TXwildlings (left panel) or wildlings and TXwildlings (right panel). The number of significantly regulated genes is indicated. The dashed line represents $q = 0.05$. **d** GSEA using Gene Ontology (Biological Processes aspect) shows the top ten most enriched gene sets in lab versus TXwildlings according to the normalized enrichment score (NES). The color indicates the significance; only significantly regulated gene sets ($q < 0.05$) are shown. **e** Cytokine and chemokine concentrations were analyzed in colon tissue by a multiplex MSD assay. Color code depicts the row-wise scaled (z-score) cytokine concentration. All cytokines that were above the detection limit in >50% of the samples are shown in the heatmap. Data are from two (**a–d**) or three (**e**) independent experiments with $n = 5$ mice per group. Median ± interquartile range (IQR) is shown. Statistical significances in (**a**) were tested by unpaired, two-tailed Mann–Whitney U tests, $p < 0.05$ is shown. Source data are provided as a Source Data file. PC Principal component. NES Normalized enrichment score.

Indeed, TXwildlings and wildlings possess 4- to 6-fold more mesenteric lymphatic tissue than lab mice (Fig. 4a), pointing towards a comparable immune status. This was further supported by the mLN transcriptome profile of TXwildlings, which closely resembled that of wildlings and differed from lab mice (Fig. 4b and Supplementary Fig. S5c). Similarly, the liver transcriptome of TXwildlings mirrored that of wildlings and differed from lab mice (Fig. 4c and Supplementary Fig. S5f). Interestingly, exposure to the natural microbiota of wildlings led to the upregulation of similar gene sets in the liver (Fig. 4d) and the colon (Fig. 3d) of TXwildlings. Further, the hepatic transcriptome analysis in TXwildlings and wildlings revealed a predominant upregulation of genes involved in immune response regulation. In contrast lab mice showed an increase in genes related to metabolic regulation (Fig. 4d). These findings are in line with increased concentrations of the T cell chemoattractants CXCL10 and CCL5 (Fig. 4e) and elevated numbers of hepatic CD8+ and CD4 + T cells in TXwildlings and wildlings compared to lab mice (Fig. 4f). Finally, we wanted to assess the systemic impact of natural gut microbiota on the lung, a gut-distant organ. In line with the overall findings, the lung transcriptome of TXwildlings resembled that of wildlings and differed significantly from lab mice (Fig. 4g and Supplementary Fig. S5i). Examining enriched gene sets in more detail revealed that the immune state in TXwildlings has shifted after microbiota transfer (Fig. 4h). This is particularly evident when displaying the significantly regulated genes of the adaptive immune response. In an unsupervised clustering, the lab mice form a separate group, while the TXwildlings and wildlings intersperse with a comparable transcription of immune response-related genes (Fig. 4i). Interestingly, the observed differences in the

transcriptome are also evident at the protein level. The cytokine and chemokine profile of TXwildlings once again mirrors that of wildlings, while differing from lab mice (Fig. 4j and Supplementary Data 2). The opposite experiment had no significant impact on wildlings transplanted with lab microbiota (Supplementary Fig. S5a, b, d, e, g, h, j–l).

Taken together, the TXsystem not only creates TXwildlings with a wildling-like phenotype in the local gut barrier site but also has the capability of inducing wildling-defining characteristics systemically with regards to a central non-lymphoid organ, lymphoid organs, and distant immunological barrier sites.

## Gut microbiota transfer drives immune maturation in TXwildlings

The systemic immunological impact of natural gut microbiota is particularly evident in the blood (cells and proteins), which serves as a vital connective system, orchestrating a complex interplay between organs and tissues throughout the body. Thus, we assessed whether the approach would also induce wildling-defining characteristics in the blood.

A flow cytometric analysis of blood CD8+ T cell subsets revealed disparities in the naive ($T_{naive}$, CD62L+CD44-), effector memory ($T_{EM}$, CD62L-CD44+), and central memory T cell ($T_{CM}$, CD62L+CD44+) compartments. In comparison to lab mice, wildlings exhibited increased proportions of antigen-experienced $T_{CM}$ and $T_{EM}$, while displaying diminished percentages of naive T cells (Fig. 5a and Supplementary Fig. S6a). CD8+ T cell subsets in TXwildlings were more similar to wildlings, with fewer naive T cells and more $T_{EM}$ cells. The levels of $T_{CM}$ remained comparable to those observed in lab mice (Fig. 5a).

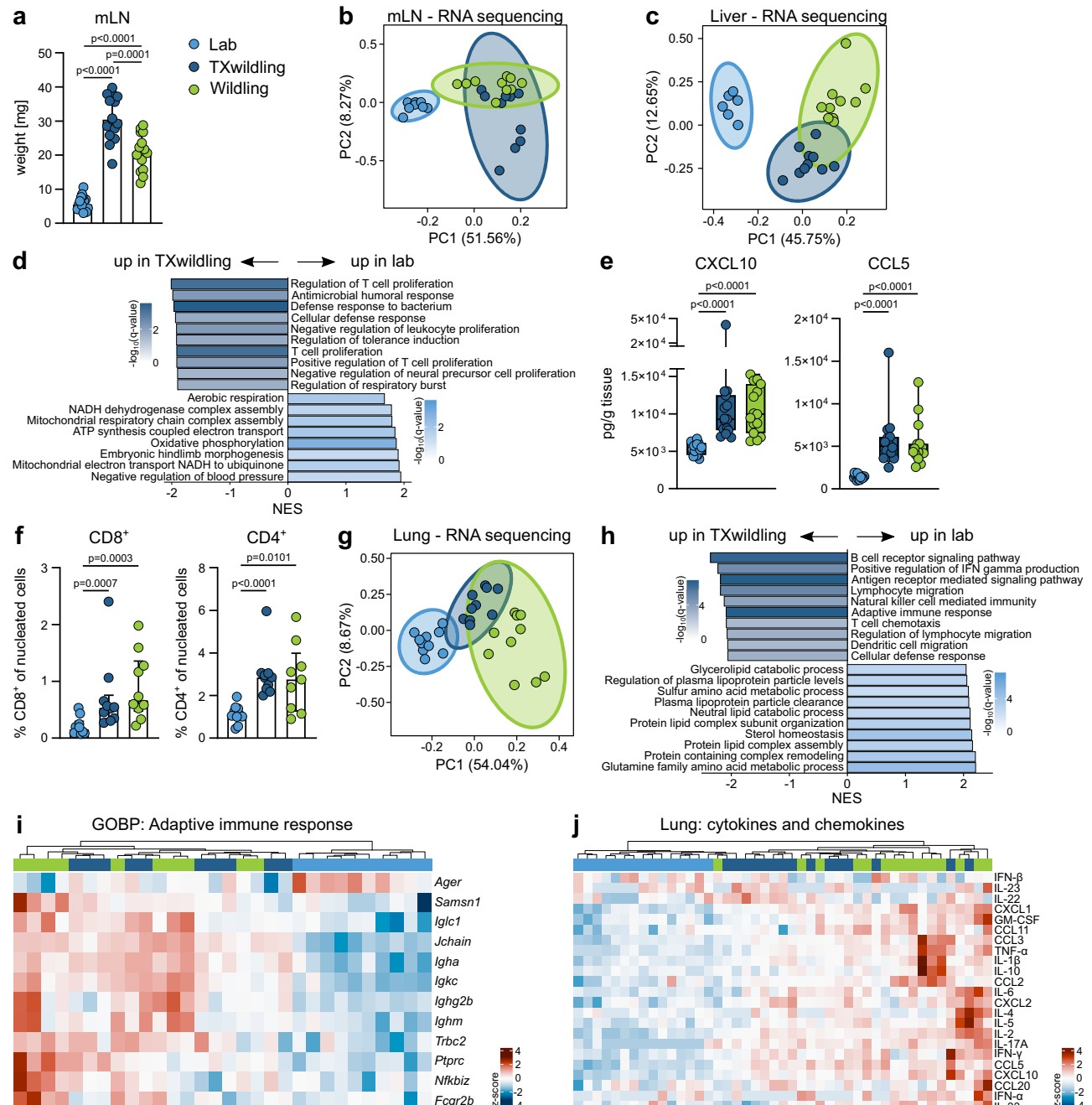

**Fig. 4 | Natural gut microbiota transplantation creates TXwildlings with systemic wildling-defining characteristics. a** Weight of mLN was measured from lab mice, TXwildlings, and wildlings. **b** mLN from lab mice, TX wildlings, and wildlings were subjected to RNA sequencing. PCA includes the genes significantly regulated between lab mice and wildlings. **c, d** Livers from mice of the three groups were subjected to RNA sequencing. **c** PCA was performed with the genes significantly regulated between lab mice and wildlings. **d** GSEA of lab mice vs. TXwildlings was conducted using Gene Ontology (Biological Processes aspect). The graph shows the top ten most enriched gene sets according to the NES. The color indicates the significance level; only gene sets significantly regulated (*q* < 0.05) are shown. **e** The concentrations of the T cell chemoattractants CXCL10 and CCL5 were measured in liver homogenates using an MSD multiplex assay. **f** The livers of mice from the three groups were subjected to multiplex immunofluorescence staining and CD4- and CD8-positive cells were quantified. **g, h** The lungs of lab mice, TXwildlings, and wildlings were subjected to RNA sequencing. **g** PCA using the genes significantly regulated between lab mice and wildlings. **h** GSEA using Gene Ontology (Biological

Processes aspect) of lab mice vs. TXwildlings shows the top ten most enriched gene sets according to the NES. The color indicates the significance, only *q* < 0.05 is shown. **i** The heatmap shows the significantly regulated genes (*q* < 0.05) of the "adaptive immune response" gene set. Color code depicts the row-wise scaled (z-score) RNA normalized expression. **j** The concentration of cytokines and chemokines was analyzed in lung tissue by a multiplex MSD assay. Color code depicts the row-wise scaled (z-score) cytokine concentration. All cytokines that were above the detection limit in >50% of the samples are shown in the heatmap. Ellipses in (**b**), (**c**), and (**g**) represent the 95% confidence interval. Data are from two (**b–d**, **f–i**) or three (**a, e**, and **j**) independent experiments with *n* = 5 per group. Median ± IQR is shown, box plots show the median, IQR, and full data spread via whiskers. Statistical significances in (**a**) were tested by unpaired, two-tailed Mann−Whitney U tests, *p* < 0.05 is shown. Source data are provided as a Source Data file. mLN Mesenteric lymph node, PC Principal component, NES Normalized enrichment score, GOBP Gene ontology – biological process.

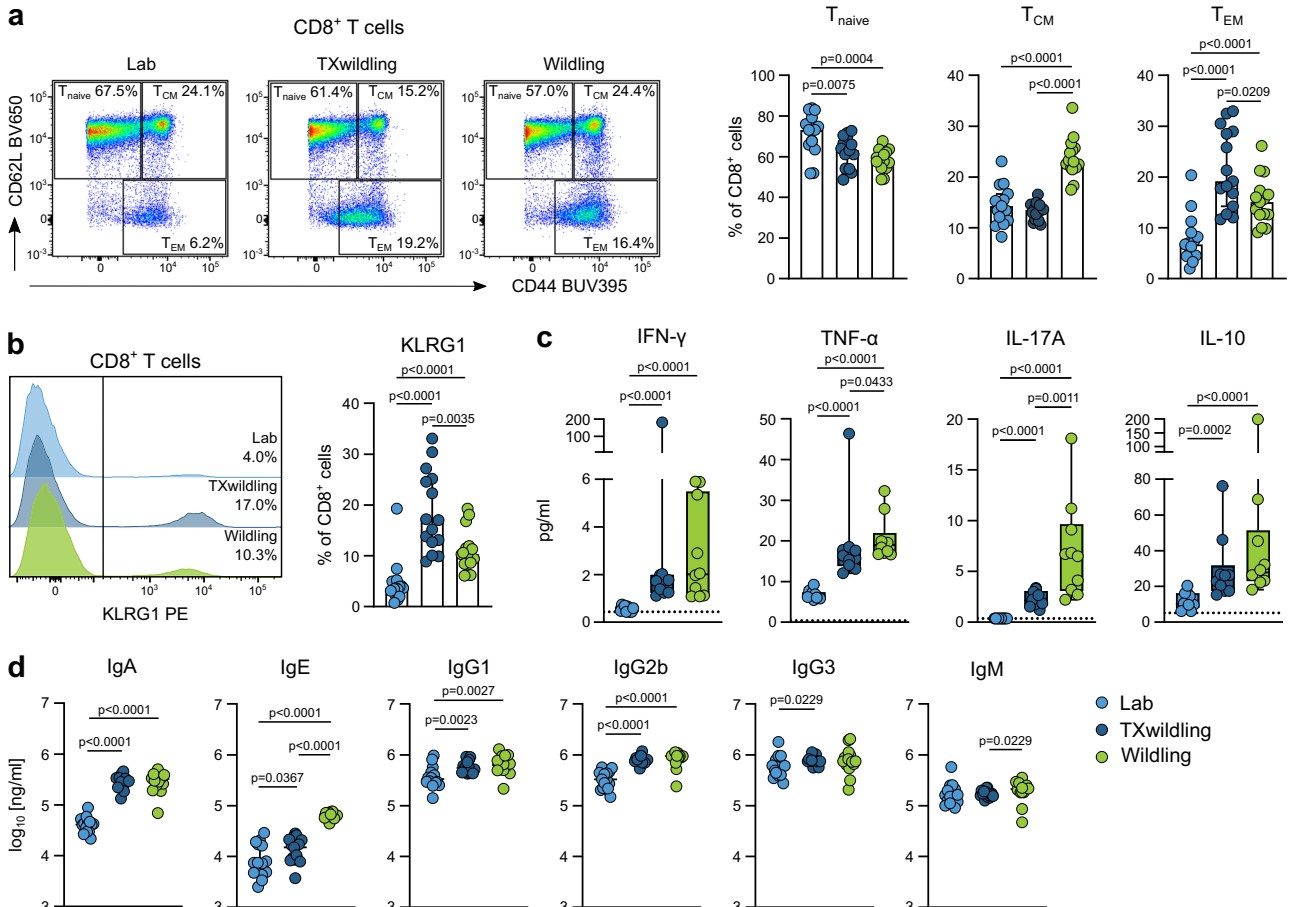

**Fig. 5 | The transfer of natural gut microbiota induces systemic immune maturation in TXwildlings. a, b** Blood cells were isolated, analyzed by flow cytometry, and gated for live CD45⁺CD3⁺CD8⁺ cells. **a** CD8⁺ T cell subsets were identified by their expression of CD44 and CD62L, as illustrated in the left panel. Specifically, $T_{naive}$ (CD62L⁺CD44⁻), $T_{CM}$ (CD62L⁺CD44⁺), and $T_{EM}$ (CD62L⁻CD44⁺) were analyzed as percentage of total CD8⁺ cells. **b** KLRG1⁺ cells were gated, as shown in the left panel, and analyzed as percentage of total CD8⁺ cells. **c** Concentrations of serum cytokines were quantified using a multiplex MSD assay.

**d** Antibody subclasses were measured in the serum of lab mice, TXwildlings, and wildlings. The shown data are from two (**c**) or three (**a, b, d**) independent experiments with n = 5 mice per group. Dashed lines in (**c**) indicate the detection limit. Median ± IQR is shown, box plots show the median, IQR, and full data spread via whiskers. Statistical significances were tested by unpaired, two-tailed Mann–Whitney U tests, p < 0.05 is shown. Source data are provided as a Source Data file.

Furthermore, the increase in effector T cells was underlined by the analysis of KLRG1⁺ CD8⁺ T cells, which were significantly more abundant in TXwildlings and wildlings compared to lab mice (Fig. 5b). Interestingly, TXwildlings exhibited even higher percentages of $T_{EM}$ and KLRG1⁺ CD8⁺ cells than wildlings (Fig. 5a, b). Moreover, wildlings and TXwildlings also exhibited increased concentrations of circulating cytokines, including IFN-γ, TNF-α, IL-17A and IL-10, in comparison to lab mice (Fig. 5c). Additionally, significant differences were observed in the circulating antibody levels between lab mice and wildlings. Wildlings showed significantly higher levels of IgA, IgE, IgG1, and IgG2b than lab mice, while IgG3 and IgM levels were comparable between the two groups (Fig. 5d). The levels of IgA, IgG1, IgG2b, and IgG3 in TXwildlings approached those of wildlings, while IgE concentrations remained at lower levels comparable to those of lab mice (Fig. 5d). Therefore, it can be concluded that TXwildlings acquired a more antigen-experienced systemic immunity comparable to that observed in wildlings. As seen in all previous datasets, the opposite experiment had no significant impact on wildlings transplanted with lab microbiota (Supplementary Fig. S6b–e).

In conclusion, the multi-organ systems analysis examining immunological barrier sites, central non-lymphoid organs, lymphoid organs, and blood shows that our transplantation approach creates TXwildlings with a systemic wildling-like phenotype.

## Similar physiology and adult human-like transcriptomes in TXwildlings and wildlings

The steady-state multi-organ systems analysis suggests that TXwildlings and wildlings share key immunological characteristics, implying that they may respond similarly to challenges, such as infectious diseases. Hence, we investigated the response of TXwildlings, lab mice, and wildlings to a gastrointestinal bacterial infection with *Citrobacter rodentium*.

Following inoculation with *C. rodentium*, laboratory mice exhibited a decline in body weight with the lowest recorded weight occurring around day 15 post-infection. In contrast to lab mice, the body weight of TXwildlings and wildlings was significantly higher and comparable throughout the infection (Fig. 6a). Additionally, TXwildlings and wildlings showed a comparable bacterial burden throughout the infection and a lower one at days four and seven compared to lab mice (Fig. 6b). This indicates that the immunological characteristics shared by TXwildlings and wildlings result in similar physiological responses.

Finally, a key characteristic of natural microbiota-based mouse models is their closer resemblance to human immunity[30,31,44,51,52]. To investigate whether TXwildlings share this mechanistic ground, we applied the analytical framework established by Reese[51] and Beura[52] to transcriptomic data from uninfected lab mice, TXwildlings, and

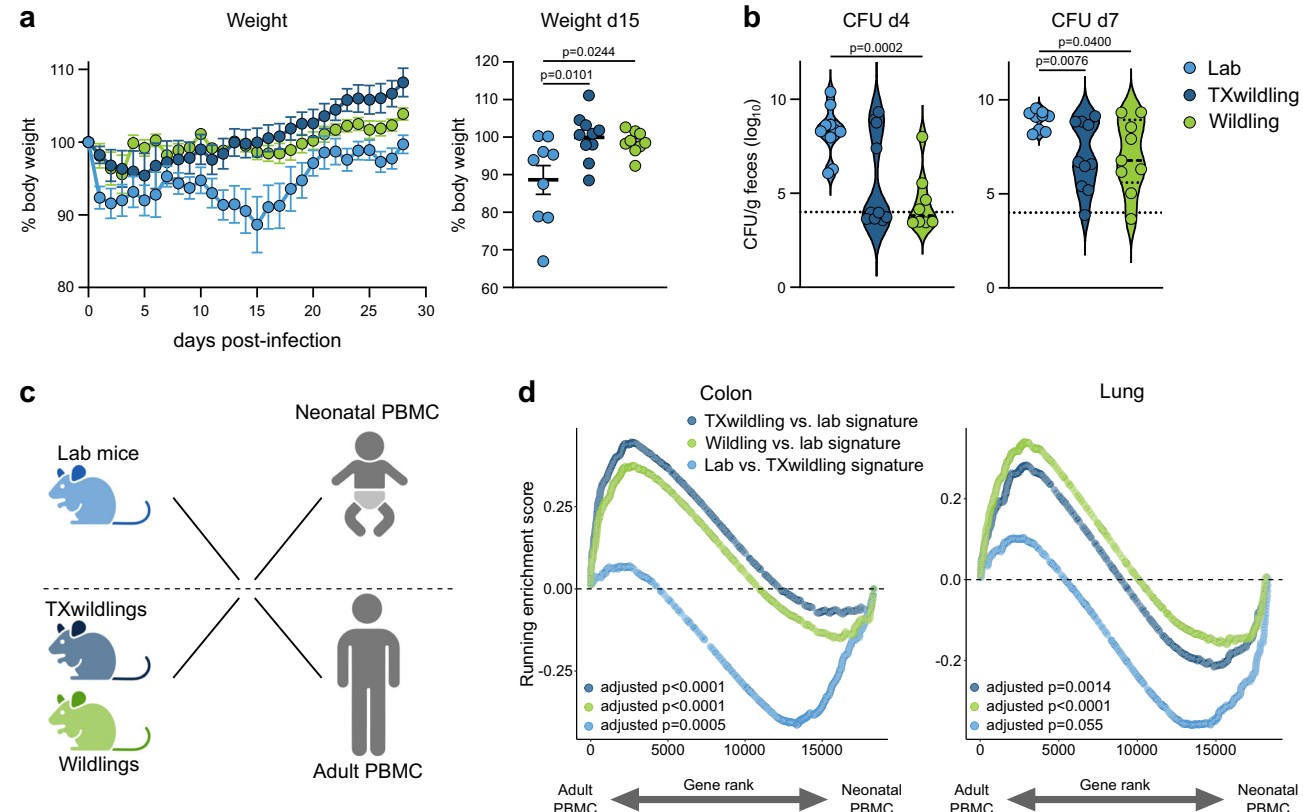

**Fig. 6 | TXwildlings and wildlings display comparable physiological responses and possess an adult human-like transcriptome.** Lab mice, TXwildlings, and wildlings were orally infected with *C. rodentium*, the body weight was analyzed until day 28 post-infection (**a**), and the fecal bacterial burden was analyzed at days 4 and 7 post-infection (**b**). **a** Body weight loss is shown over time and for the greatest significant weight difference on day 15 post-infection. **c, d** GSEA among the indicated mouse comparisons of uninfected lab mice, TXwildlings and wildlings compared to a ranked gene list derived from healthy adult vs. neonatal PBMCs.

Gene signatures include the top 500 significantly differently expressed genes of colon or lung tissues of TXwildlings vs. lab mice (dark blue), wildling vs. lab mice (green) or lab mice vs. TXwildlings (light blue). Data are from two independent experiments with *n* = 5 per group. Dashed lines in (**b**) indicate the detection limit. Mean ± SEM is shown. Statistical significances in (**a**), right panel and (**b**) were tested by unpaired, two-tailed Mann–Whitney U tests, *p* < 0.05 is shown. Figure in (**c**) was created in BioRender. Bruno, P. (2025) https://BioRender.com/2y4432p. Source data are provided as a Source Data file. CFU: Colony-forming units.

wildlings. The GSEA compared a ranked gene list of PBMCs isolated from adult and neonatal humans to the list of the top 500 significantly differently regulated genes in each group (Fig. 6c)[62]. The adult human transcriptome was significantly enriched in the colon and lung transcriptome of TXwildlings and wildlings compared to lab mice, whereas lab mice showed a greater similarity to neonatal humans than TXwildlings or wildlings (Fig. 6d).

In summary, we have shown that TXwildlings and wildlings share key immunological features, display similar physiological responses, and better represent adult humans than lab mice.

## Discussion

Lab mice have been essential in shaping our understanding of mammalian physiology. However, their low resilience and facility-specific microbiota result in phenotypes that vary across research institutions. These differences influence immune and non-immune cell functions, causing systemic physiological changes that generate irreproducible and contradictory data[15,21,26–29]. Hence, to increase the reproducibility of mouse data, the microbiota must be harmonized with a standard that embodies the needed complexity and resilience to create robust models across different institutions. Natural microbiota possess the required characteristics to overcome this global reproducibility crisis[41]. However, developing natural microbiota-based models requires substantial funding, expertise, and specialized infrastructure, limiting their accessibility for widespread adoption. Thus, our study aimed to develop and validate a standardizable, feasible, and

controlled TXsystem capable of converting lab mice into natural microbiota-based models.

The microbiota profiling data demonstrated that the natural gut microbiota of wildlings swiftly outcompeted the microbiota of fully colonized adult lab mice, despite using just a single oral gavage and a numerical disadvantage in a 1:4 co-housing setup. Remarkably, out-competition occurred in all globally leading vendors, underscoring the superior co-evolutionary adaptation of natural microbiota to the gut niche[63,64], arguing for a generalizable phenomenon. To add functional information to the taxonomic 16S rRNA gene profiling, we performed a global metabolomics analysis of cecal content assessing the function of the gut microbiota[65]. We found that the microbiota of TXwildlings not only mimicked the community structure of wildling microbiota but also closely resembled their functional capability.

In addition to differences in the microbiota, wildlings and lab mice are known to differ significantly in various aspects, such as anatomical features and their immune systems[31]. To investigate if TXwildlings would adopt these defining characteristics of wildlings, we conducted a multi-organ systems analysis examining immunological barrier sites, central non-lymphoid organs, lymphoid organs, and blood. Our findings show that TXwildlings share anatomical features with wildlings, such as the size of the cecum and mesenteric lymph nodes compartment. Notably, transcriptomic, cellular, and cytokine analyses show that TXwildlings share key immunological traits with wildlings while being significantly different from lab mice. This is not only true for the local barrier site but also for a central non-lymphoid organ, lymphoid

organs, and a distant barrier site. Further, TXwildlings possess a mature immune system, characterized by an elevated number of effector and memory T cells and increased cytokine and immunoglobulin levels. However, it is important to emphasize that the natural microbiota of TXwildlings are limited to the gut, while wildlings are colonized with wild mouse microbiota across all epithelial barrier sites, including the skin, vagina, and lung[31]. Additionally, TXwildlings are not exposed to natural microbiota throughout all developmental stages, potentially overlooking effects during ontogeny[31,34]. Consequently, we cannot exclude the possibility of other microbiota niches and/or ontogenetic effects contributing to the differences in physiological and immunological traits observed between TXwildlings and wildlings. Nevertheless, our multi-organ systems analysis suggests that TXwildlings and wildlings share key immunological characteristics, implying that they may respond similarly to physiological challenges, such as infectious diseases. Indeed, TXwildlings and wildlings exhibited comparable body weight and bacterial burden during infection with *C. rodentium*. Notably, this phenotype cannot be attributed to differences in the presence of segmented filamentous bacteria (SFB), as TXwildlings, wildlings, and Taconic lab mice were all colonized with SFB. This finding is particularly important, as microbiota-mediated protective phenotypes have also been reported in other natural microbiota-based models in the context of infectious diseases[30,52,66] and cancer[30,67]. These results suggest that TXwildlings may provide a valuable platform for uncovering novel disease-relevant mechanisms that are not detectable in laboratory mice[50].

Finally, all natural microbiota-based mouse models share a common mechanistic ground: their complex microbiota and/or microbial exposure result in mice with mature, experienced, and trained immune systems that more closely resemble adult human immunity[30,31,44,51,52]. To investigate whether TXwildlings share this mechanistic ground, we applied the analytical framework established by Reese[51] and Beura[52]. Our analysis revealed that the transcriptome of TXwildlings closely resembles that of wildlings and better represents adult humans than lab mice. Since wildlings have been shown to phenocopy human responses and could have prevented failed clinical trials[31,68,69], it is plausible to hypothesize that TXwildlings may similarly enhance the safety and efficacy of translational research efforts, as demonstrated with wildlings.

Taken together, microbiota standardization addresses the reproducibility crisis by resolving the issue of divergent microbiota across institutions, though no suitable candidate has been identified to date[41]. In this work, we presented an evidence-based rationale and proof-of-concept study to use natural gut microbiota as candidate for standardization due to their key attributes mentioned in the introduction (I-V). We developed and validated the TXwildling system – a scalable, standardizable, and controlled approach to make natural microbiota-based models globally accessible on various genetic backgrounds. Although this needs to be further validated and tested in different locations, we anticipate that the widespread adoption of TXwildlings may enable the discovery of novel treatments, enhance reproducibility, reduce research costs, and improve the safety and success of translational efforts, ultimately advancing human health.

## Methods
### Mice
Laboratory mice used in this study were C57BL/6NTac (MPF) ordered from Taconic, Denmark if not indicated otherwise. For one experiment, laboratory mice with SPF microbiota were purchased from The Jackson Laboratory (C57BL/6J), Charles River (C57BL/6NCrl), Envigo (C57BL/6JRccHsd), and Janvier Labs (C57BL/6JRj). Wildling C57BL/6NTac mice were created by embryo transfer of laboratory mouse into pseudopregnant wild mice[31] and bred locally. All mice were housed at the Medical Center – University of Freiburg, Germany. Throughout the entire experiment, mice were handled under BSL1 conditions and were housed under a 12:12 light:dark cycle (room temperature 20–24 °C, humidity 45%-65%, air exchange rate of 15 times per hour) in a Greeline IVC system from Tecniplast inside of autoclaved microisolator cages (GM500) with autoclaved rodent chow (GRANOVIT AG, KLIBA NAFAG, 3437.PX.L15) and autoclaved tap water ad libitum, 1x autoclaved sizzle pad 8 g (ssniff), 1x autoclaved play tunnel (ssniff), 1x autoclaved smart home, and Aspen wood chip bedding (ssniff). Due to the 1:4 co-housing setting, only female mice were used in all experiments. Experiments were performed in accordance with the guidelines of the Federation for Laboratory Animal Science Associations and the national animal welfare body. They were in compliance with the German animal protection law and were approved by the animal welfare committee of the Regierungspräsidium Freiburg (permit G-21/030). Mice used for experiments were 12–16 weeks old.

### Pathogen testing
Fecal pellets and blood dried on Opti-Spot card were collected from wildlings and TXwildlings according to the manufacturer's sampling guidelines (IDEXX BioAnalytics). Serologic analysis using the Global Serology panel was performed by IDEXX BioAnalytics on individual samples. PCR analysis was performed on pooled samples using the Mouse 3R Quarantine Annual SOPF (PCR) panels and PCR for MCMV and Hantaan. A microorganism was considered present if it was identified through at least one of the two independent methods.

For C57BL/6NTac lab mice, the hygiene standard was assessed and reported by the mouse vendor (Taconic Biosciences).

### Gut microbiome transplantation
In general, the ileocecal microbial communities from terminal ileum and cecum of wildlings and SPF mice were transferred by a single oral gavage into only one out of five mice per cage. To more effectively encompass the microbial diversity of the wildling colony, five wildlings were euthanized with $CO_2$ and subsequently harvested. The fur was disinfected with Bacillol, their abdominal cavity was opened under sterile conditions using sterile equipment and autoclaved surgical instruments in a biosafety cabinet class 2. Two separate sets of instruments were used, one set for the skin and one set for the intraabdominal preparation.

To minimize exposure to oxygen, the terminal ilea and ceca of all 5 donor mice were extracted from the peritoneal cavity without opening them and placed in a sterile bacteriological petri dish (Greiner Bio-One cat. #633181). Subsequently, they were opened and manually extruded along the cephalocaudal axis to collect the fecal material into a sterile 100 µm cell strainer (Corning Falcon cat. #352360) placed inside of a sterile bacteriological petri dish. The collected fecal material was weighed and then diluted at a 1:2 ratio (g collected material: ml fluid) in sterile filtered cryoprotectant (PBS (Panbiotech, P04-361000) containing 0.1% L-cysteine (Sigma-Aldrich, C7352) and 20% glycerol (Sigma-Aldrich, G2025)). The resulting suspension was subsequently mashed through the 100 µm cell strainer to remove all insoluble particles, 1 ml aliquots were prepared into 2 mL externally threaded plastic screw-capped cryo-vials (Azenta Life Sciences, BCS-2502), subjected to controlled freezing with 1 °C per minute (Azenta Life Sciences CoolCell LX Cell, BCS-405) and stored in a −80 °C mechanical freezer. Prior to inoculation, the samples were thawed in a 37 °C water bath and used for the inoculation within 10 min. Using a sterile syringe (B. Braun, 9161708 V) and gavage needle (Merck Cadence Science, CAD9921-100EA), one out of five mice was inoculated one time with a total of 100 µl of this suspension (equals 50 mg of fecal material) via oral gavage. Harvest, preservation and transplantation of microbiota from SPF mice was performed in an identical fashion. After inoculation, all experimental mice were placed into clean cages followed by weekly cage changes.

## C. rodentium infection

For the infection experiments the streptomycin-resistant *C. rodentium* strain DBS100 (provided by Bruce Vallance) was used. Bacteria were grown on LB agar plates with streptomycin (100 µg/ml) at 37 °C overnight and in LB medium with shaking (200 rpm) at 37 °C overnight. Bacterial suspension was diluted 1:50 and mice were orally gavaged with $10^7$ CFU of *C. rodentium* in 100 µl. Weight of the mice was monitored daily and feces were collected at the indicated time points for measuring the bacterial burden. For this, feces were collected and homogenized in 1 ml PBS with 1.5 mm ceramic beads (Biolabproducts, 21-25-ZY16.5) three times for 1 min at 5 m/s using a Bead Ruptor Elite (Omni International, SKU 19-040E). Serial dilutions of homogenized fecal pellets were plated on LB agar plates with streptomycin (100 µg/ml) and cultured at 37 °C overnight. CFU were counted and calculated by normalizing to the weight of each sample.

## 16S rRNA gene amplicon sequencing

Fecal pellets were collected from all mice at the indicated time points, snap frozen, and stored at −80 °C. First, genomic DNA was extracted using the ZymoBIOMICS 96 MagBead DNA Kit according to the manufacturer's instructions. 16S rRNA gene amplification of the V4 region (F515/R806) was performed according to an established protocol previously described[70]. Briefly, DNA was normalized to 25 ng/µl and used for sequencing PCR with unique 12-base Golay barcodes incorporated via specific primers (obtained from Sigma). PCR was performed using Q5 polymerase (NewEnglandBiolabs) in triplicates for each sample, using PCR conditions of initial denaturation for 30 s at 98 °C, followed by 25 cycles (10 s at 98 °C, 20 s at 55 °C, and 20 s at 72 °C). After pooling and normalization to 10 nM, PCR amplicons were sequenced on an Illumina MiSeq platform via 250 bp paired-end sequencing (PE250). Resulting raw reads were demultiplexed by idmp (https://github.com/yhwu/idemp) according to the given barcodes[70]. Libraries were processed including merging the paired end reads, filtering the low-quality sequences, dereplication to find unique sequences, singleton removal, denoising, and chimera checking using the USEARCH pipeline version 11.0.667[71]. In brief, reads merged by fastq_mergepairs command (parameters: maxdiffs 30, pctid 70, min-mergelen 200, maxmergelen 400), filtered for low quality with fastq_filter (maxee 1) and singletons using fastx_uniques command (minuniquesize 2). We run UPARSE algorithm to make 97% OTUs and filter chimeras calling, following the amplicon quantification using usearch_global command (strand plus, id 0.97, maxaccepts 10, top_hit_only, maxrejects 250). Taxonomic assignment was conducted by Constax (classifiers: rdp, sintax, blast) using the GreenGenes2 database[72] and summarizing into biom-file for the visualization in phyloseq[73] and downstream analysis.

## ITS sequencing

DNA extraction and ITS sequencing were performed by Zymo Research Europe, Freiburg, Germany. DNA was extracted using the ZymoBIOMICS®−96-MagBead DNA Kit (Zymo Research) according to the manufacturer's instructions. The DNA samples were prepared for targeted sequencing with the Microbiome Analysis Services ITS2 Primer Set. The sequencing library was prepared using an innovative library preparation process in which PCR reactions were performed in real-time PCR machines to control cycles and therefore limit PCR chimera formation. The final PCR products were quantified with qPCR fluorescence readings and pooled together based on equal molarity. The final pooled library was cleaned up with the Select-a-Size DNA Clean & Concentrator™ (Zymo Research), then quantified with TapeStation® (Agilent Technologies) and Qubit® (Thermo Fisher Scientific). The final library was sequenced on Illumina® NextSeq™ 1000 with a P1 XLEAP-SBS™ Reagent Kit (600 Cycles). The sequencing was performed with 40% PhiX spike-in. A quantitative real-time PCR was set up with a standard curve. The standard curve was made with plasmid DNA containing one copy of the 16S gene and one copy of the fungal ITS2 region prepared in 10-fold serial dilutions. The primers used were the same as those used in Targeted Library Preparation. The equation generated by the plasmid DNA standard curve was used to calculate the number of gene copies in the reaction for each sample. The PCR input volume (4 µl) was used to calculate the number of gene copies per microliter in each DNA sample.

Libraries were processed including merging the paired end reads, filtering the low-quality sequences, dereplication to find unique sequence, singleton removal, denoising, and chimera checking using the USEARCH pipeline version 11.0.667[71]. In brief, reads merged by fastq_mergepairs command (parameters: maxdiffs 30, pctid 70, min-mergelen 300, maxmergelen 400), filtered for low quality with fastq_filter (maxee 1) and singletons using fastx_uniques command (minuniquesize 2).

To predict biological sequences (ASV,zOTUs) and filter chimeras we the unoise3 command (minsize 10, unoise_alpha 2), following the amplicon quantification using usearch_global command (strand plus, id 0.97, maxaccepts 10, top_hit_only, maxrejects 250).

Taxonomic assignment was conducted by Constax (classifiers: rdp, sintax, blast) using the UNITE database[72,74] and summarizing into biom-file for the visualization in phyloseq[73] and downstream analysis.

## Shotgun sequencing

Metagenomic libraries were prepared using the Illumina DNA PCR-Free Library Kit and IDT for Illumina DNA/RNA UD Indexes with previously isolated community DNA. Library preparation followed by the manufacturer's protocol and quantification of library concentrations was performed using the Qubit ssDNA Assay Kit, followed by additional quantification with the KAPA Library Quantification Kit for Illumina. Sequencing was carried out on the NovaSeq S4 PE150 platform with a depth of 25 million reads per sample. Read were trimmed for low adaptor sequences and low quality and filtered for phiX and mouse host reads using the bbmap-software-tools[75].

To determine non-bacterial species, we use a denovo-assembly and contig-classification method. Libraries were assembled using megahit[76]. Resulting Contings were filtered to lengths of minimum 1000 bp and reads of all libraries were mapped via bwa2mem. Contigs were taxonomic classified via CAT[77]. Mapped reads were normalized to TPM and summarized to different taxa.

## Metabolomics

Blood was collected by cardiac puncture into EDTA-coated monovettes (Sarstedt, 06.1660.100). After centrifugation ($2300 \times g$, 12 min, 20 °C), plasma was collected and frozen at −80 °C. Global untargeted metabolomics, were performed by Metabolon as described elsewhere in detail[78]. Briefly, samples were prepared including precipitation and removal of protein with methanol and subsequent removal of organic solvent. To ensure broad coverage, the extracts were measured using several mass spectrometry platforms differing in their physicochemical properties such as mass, charge, and ionization behavior. For quality assurance, several controls were used, including a pooled matrix sample, a blank control, and cocktails of QC standards. Metabolites were identified by comparing all molecules in the proprietary library[79,80] regarding retention time/index, mass/charge ratio, and fragmentation data. Peaks were quantified by area-under-the-curve and values were batch-normalized, normalized to the median, and log-transformed.

## RNA sequencing

Organ pieces for RNA sequencing were snap frozen in liquid nitrogen and stored at −80 °C until further use. RNA isolation, library preparation, and 3' sequencing were performed by Qiagen. Briefly RNeasy Kit was used according to the manufacturer's instructions for RNA extraction and 7 ng of RNA were converted to cDNA using

the QUAseq UPX 3' Transcriptome Kit. This resulted in tagging each RNA molecule with a unique molecular identifier (UMI) and tagging each sample with a unique ID. After amplification of the cDNA and library purification, it was quality controlled by capillary electrophoresis using Agilent DNA 7500 Chip). For sequencing, the libraries were pooled and sequenced on a NovaSeq sequencing instrument according to the manufacturer's instructions. The raw sequencing reads were demultiplexed for the sample indices using the CLC Genomics Workbench 20.0.4 "Demultiplex QIAseq UPX 3' reads" tool and bcl2fastq2 software was used to generate FASTQ files for each sample. UMI of first 12 nucleotides, serving as unique RNA molecule identifiers, were extracted from the sequencing reads using UMI-tools "extract" (version 1.1.4)[81]. Subsequent read trimming was performed with fastp (version 0.23.4)[82] using the parameters: -q 20, --trim_tail1 1, --trim_poly_x, and --poly_x_min_len 10. The trimmed reads were aligned to the mouse reference genome (mm10) utilizing STAR (version 2.7.11a)[83]. Post-alignment, deduplication of reads based on their UMI barcodes was executed using UMI-tools "dedup". Gene-level quantification of reads was conducted with htseq (version 2.0.3)[84]. For downstream analysis, the R software environment (version 4.3.2) was employed. Raw read counts were normalized accounting for library size and the TMM (Trimmed Mean of M-values) normalization factor. Differential expression analysis was conducted using the edgeR[85] package within R. Gene-set enrichment analysis was performed using the clusterProfiler[86] package in R, referencing the MSigDB mouse database[87] and custom gene-sets derived from Votavova et al.[62] using mouse-to-human ortholog mapping from homologene R package. In both analyses, significance was determined by an adjusted p-value (Benjamini-Hochberg correction) below 0.05.

## Histology

Organs for histological staining were fixed in 4% PFA (Morphisto, 11762) overnight and paraffin-embedded. Slides were de-paraffinized, rehydrated and antigen retrieval was conducted by pressure cooking in citrate buffer (pH 6) for 5 min. Opal 4-Color Anti-Rabbit Manual IHC Kit (Akoya Biosciences, NEL840001KT) was used for staining of CD45 (Abcam, ab10558, dilution 1:200, incubation 90 min at RT), CD4 (Abcam, ab183685, dilution 1:500, incubation 30 min at RT) and CD8α (Cell Signaling, 98941, dilution 1:200, incubation 30 min at RT) according to the manufacturer's instructions.

Pictures were acquired using an Akoya Phenocycler Fusion 1.0.5. microscope (Akoya Biosciences) and quality-controlled using Phenochart (version 1.2.0, Akoya Biosciences). Three regions of interest (926 μm × 695 μm) were selected per sample and exported using inForm software (version 2.8.0, Akoya Biosciences). Exported pictures were analyzed using QuPath (version 0.3.2)[88] and StarDist algorithm for cell segmentation[89]. Cells were quantified and segmented using DAPI staining. Individual classifiers were trained for CD4 and CD8 expression each using co-expression of DAPI and the respective maker. After training the algorithm, each picture was analyzed using multi-sample analysis.

## Antibody isotyping

Blood was collected by cardiac puncture into serum gel tubes (Sarstedt, 411378005) and centrifuged after clotting (10000 × g, 5 min, 20 °C). Serum was collected and stored at −80 °C until further use. Prior to antibody measurements, serum was centrifuged (20,000 × g, 10 min, 4 °C) to remove any precipitates. For antibody isotyping, a Mouse Isotyping Panel 1 Kit (MSD, K15183B-1) and an IgE ELISA (Invitrogen, 88-50460) were used according to the manufacturer's instructions.

## Cytokine measurement

Organs for cytokine analysis were snap frozen and stored at −80 °C until further processing. Frozen organs were homogenized in Tissue Lysing buffer (1x Tris Lysis buffer, 2% Phosphatase Inhibitor I, 2% Phosphatase Inhibitor II, 2% Protease Inhibitor solution (MSD, R70AA-1), 2 mM PMSF (Roche, 10837091001)) with a 6.35 mm metal bead (Biolabproducts, 21-25-NS64.10) using 4 cycles at 5 m/s for 20 s on a Bead Ruptor 24 Elite (Biolabproducts). The organ homogenates were centrifuged (20,000 × g, 5 min, 4 °C), the supernatant collected, frozen and stored at −80 °C. For cytokine and chemokine analysis, a custom 35-plex U-plex assay (MSD) was used according to the manufacturer's instruction (CCL11, CCL2, CCL20, CCL3, CCL5, CXCL1, CXCL10, CXCL2, GM-CSF, IFN-α, IFN-β, IFN-γ, IL-10, IL-12p70, IL-13, IL-15, IL-16, IL-17A, IL-17A/F, IL-17C, IL-17E/IL-25, IL-17F, IL-1β, IL-2, IL-21, IL-22, IL-23, IL-27p28/IL-30, IL-31, IL-33, IL-4, IL-5, IL-6, IL-9, TNF-α). The analysis was done using a QuickPlex SQ 120 instrument (MSD) and DISCOVERY WORK-BENCH version 4.0 software (MSD).

## Flow cytometry analysis of blood cells

Blood was collected by cardiac puncture into EDTA-coated monovettes (Sarstedt, 06.1660.100). For erythrocyte lysis, the blood was added to 10 ml of 1x RBC Lysis Buffer (BioLegend, 420301) and incubated for 4 min. After addition of 40 ml of PBS, the cell suspension was centrifuged (336 × g, 12 min, 4 °C) and the cells were counted. A total of 2 × 10⁶ cells per sample were seeded and incubated with viability dye eFluor780 (Life Technologies GmbH, 65086518) for 15 min at 4 °C, followed by incubation with anti-CD16/CD32 antibody (BD, 553141) for 15 min at 4 °C and incubation with fluorophore-labeled surface antibodies (20 min, 4 °C). A detailed list with information about the antibodies is available in Supplementary Table 3. After washing the cells, they were fixed with 2% PFA (Morphisto, 11762) and measured on a Fortessa LSR (BD).

## Statistics & reproducibility

Statistical analyses were performed with Prism version 10.2.3 (GraphPad Software) unless otherwise stated. Two-tailed nonparametric t tests were used (Mann–Whitney U test for two groups).

Sample size for mouse studies was chosen according to institutional directives and in accordance with the 3Rs (Replacement, Reduction and Refinement) guiding principles underpinning the humane use of animals in research, but no statistical analyses were performed to predetermine the sample sizes. The sample size of 10 or 15 animals in most groups results from 5 mice being in one experiment, representing two or three experiments, respectively. Data that failed quality controls were excluded. All attempts at replication were successful, with multiple mice in each group. Some animal experiments were not replicated, but all experimental groups consisted of at least 5 animals with similar results. Also, we observed similar results with mice from different vendor, indicating well-replicable results. Mice were randomly allocated to different treatments. Blinding was not applied due to hygiene standards.

## Reporting summary

Further information on research design is available in the Nature Portfolio Reporting Summary linked to this article.

## Data availability

The RNA sequencing data generated in this study have been deposited in the Gene Expression Omnibus (GEO) database under accession code GSE287225. The RNA sequencing data used for creating Fig. 6d are available in the GEO database under accession code GSE27272. The 16S rRNA and ITS2 gene amplicon and shotgun sequencing data generated in this study have been deposited in the BioProject database under accession code PRJEB84922. The metabolomics data generated in this study (Fig. 2) are provided in Supplementary Data 1. The cytokine data generated in this study (Figs. 3e and 4j) are provided in Supplementary Data 2. Source data are provided with this paper.

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

## Acknowledgements

The authors thank the Lighthouse Core Facility (funded by the Deutsche Forschungsgemeinschaft (DFG, German Research Foundation) – 450392965) for their excellent assistance with the acquisition of histology pictures and Bruce Vallance for providing the *C. rodentium* DBS 100 stock. We authors acknowledge the support of Achim Gronow for support of microbiota analysis. This work was funded by the Deutsche Forschungsgemeinschaft (DFG, German Research Foundation) SFB1160 IMPATH (project ID 256073931, B10N to S.P.R., Z01 to C.S., Z02 to M.B.), the DFG Emmy Noether-Program RO 6247/1-1 (project ID 446316360 to S.P.R.), the TRR 359 PILOT (project ID 491676693, A07 to S.P.R., Z01 to M.B.), the SFB/TRR167 (project ID 259373024, Z01 to M.B.), the SFB1453

(project ID 431984000, P12 to C.S., S1 to M.B.), SFB1479 (project ID: 441891347- S1 to M.B.), the SFB/TRR353 (project ID 471011418, SP02 to M.B.), the FOR 5476 UcarE (project ID 493802833-P7 to M.B.), SCHE 2092/4-1 (RP9, CP2, CP3) (project IDs 241702976 and 438496892 to C.S.), the Heisenberg program (project ID 501370692 to C.S.), and Germany's Excellence Strategy - EXC 2155 (project number 390874280 to T.S.). We also acknowledge funding from the German Federal Ministry of Education and Research (BMBF) within the Medical Informatics Funding Scheme PM4Onco–FKZ 01ZZ2322A (M.B.) and EkoEstMed–FKZ 01ZZ2015 (G.A.), the European Research Council (CoG 865466 to T.S.). and the Wilhelm Sander-Stiftung (project ID 2023.010.1 to C.S.).

## Author contributions

Conceptualization: B.H. and S.P.R. Methodology: S.R., S.v.Z., A.M.M., B.Z., M.R., S.P.R. Software: S.R, L.O., T.R.L., G.A., B.H. Formal Analysis: S.R, L.O., T.R.L., G.A., B.H. Investigation: S.R., S.v.Z., A.M.M., L.O., A.M.G., B.Z., K.G. Data Curation: S.R., S.v.Z., A.M.M., L.O., T.R.L., G.A., B.H. Writing – Original Draft: S.R., P.B., B.H., S.P.R. Writing – Review & Editing: S.R., P.B., B.H., S.P.R. Visualization: S.R., S.v.Z., G.A., B.H. Supervision: S.R., C.S., M.B., T.S., S.P.R. Project Administration: S.P.R. Funding Acquisition: C.S., M.B., T.S., S.P.R.

## Funding

## Competing interests

S.P.R. has no conflict of interest and discloses that NIDDK granted a license on the WildR mice[30] to Taconic Biosciences. The other authors declare no competing interests.
