## [Transparent Peer Review file · Nature Communications]

Laboratory mice engrafted with natural gut microbiota possess a wildling-like phenotype

Corresponding Author: Professor Stephan Rosshart

Version 0:

Reviewer comments:

Reviewer #1

(Remarks to the Author)

The manuscript was easy to read and presents convincing data on the relevance of the new mouse model presented. I have only one major comment and additional minor points.

The author emphasises both the need for the community to have additional, more representative mouse models available and the need to provide standardised approaches for doing so. In contrast, I felt that the methods were not sufficiently described. If I understand one of the authors' aims correctly, namely to help the community by paving the way for the use of a model that is better than conventional mice and easier to use than wildlings, I would argue that it is not currently possible for readers to understand all the technical aspects behind the model and potentially reproduce the work. The methods around the preparation of the gut content stocks (e.g. sterile conditions? anaerobic conditions? volume of aliquots? etc.) and, perhaps more importantly, the gavage procedure (e.g. disposable aliquots, waiting time between melting and gavage, etc.). Diet information is also needed. Access to wildlings samples may also need to be clearer in the manuscript. These are only examples, and I urge the authors to think carefully and provide all the technical details necessary to reproduce this nice work.

The mice used in the experiments were relatively old (12-16 weeks) considering e.g. the development of immune responses; I think this deserves a comment / explanation, perhaps in the discussion.

I found the parts about similarity to the human situation less relevant. While wildlings have been used in the past to successfully refute previous preclinical data obtained using standard laboratory mice, the main point is to enrich the range of models available. A devil's advocate could probably list several negative aspects of TXwildlings. The authors may wish to provide a more balanced view.

L68: "the metagenome, the combination of the microbiome and the host genome"; this is incorrect;
<https://doi.org/10.1186/s40168-020-00875-0>; <https://doi.org/10.1186/s40168-015-0094-5>

L74: naturally coevolved microbes, not microbiota; the term microbes may be more appropriate also in other parts of the manuscript?

The terms "bacterial microbiome" (L76) or "bacterial gut microbiome" (L675), perhaps also elsewhere in the manuscript, are inaccurate.

L211: While there were (always past tense in the methods and results)

Reviewer #2

(Remarks to the Author)

In this study Runge et al. demonstrate some interesting characteristics of a wildling mouse microbiome transplanted into laboratory mice (referred to as TXwildling mice in the paper). The authors show that the gut microbiome of wildling mice consistently invades and apparently replaces the microbiota of co-housed SPF cagemates, regardless of the ratio between

TXwildling and SPF mice. The paper also comprehensively characterises immune phenotypes of TXwildling mice both locally in the gut but also at distant body sites such as liver and lung, showing them to be rather similar to those of wildlings from which their microbiome is derived, and also more comparable to a mature human than that of SPF lab mice. Moreover, both the TXwildling and wildling transcriptomes (colon and lung) resembled that of adult human more closely than did that of lab mice. These results suggest that compared to the typical microbiome of SPF lab mice, the TXwildling microbiome may provide a more stable platform which better recapitulates the phenotypes of wild mice and adult humans, for use in biomedical research. The authors argue that since the creation of natural microbiota-based mouse models come with significant logistical hurdles, this TXwildling system presents a feasible alternative to convert standard lab mice to natural microbiota-based models.

We see significant value in this work, and agree that such a natural mouse model with a resilient wild-derived microbiome holds real value for mouse research. However, we do have some concerns with the work as presented that we think need addressing to maximise its impact.

1) Value of TXwildlings as a standardized wild mouse microbiome model

Throughout the text the authors highlight that the TXwildling system provides a stable wildling-sourced microbiome which they anticipate could provide a useful tool for standardizing lab mouse microbiota in future studies. However, in light of the study's scope and the data presented, currently this seems a rather strong claim, for reasons expanded upon below.

The authors state (lines 121-122) that natural microbiota-based models demand significant funding, expertise and specialised infrastructure. But that widespread adoption of TXwildlings could make them more globally accessible (line 463). In order to assess how TXwildlings alleviate these hurdles, it would be useful to be more explicit what exactly these challenges are and how TXwildlings overcomes them. Isn't one of the major challenges of natural microbiota models the potential presence of pathogens, and thus the need to house them outside standard lab animal facilities? If so, it is unclear how TXwildlings would overcome this at present, as no pathogen screening results have been presented (and it is not clear in the original wildling paper what might transmit via gut contents). We would therefore suggest including comprehensive pathogen screening data (and ideally metagenomic data to assess viruses that may not have dedicated assays) to support that they are pathogen-free, or otherwise modify this claim.

In line with this, we are unsure why the authors chose to use wildling microbiome as the source microbiome for this study and to recommend (line 112) as opposed to the WildR microbiome previously generated by this research group (Rosshart et al., 2017), which has already been pathogen-screened and is now a commercially-available 'SPF' wild mouse microbiome model. This would seem to be more readily 'safe' and adoptable by others keen to naturalise their mouse models' microbiota through co-housing.

The manuscript largely implies the bacterial microbiota as being the key factor driving different immune phenotypes (largely implicitly, by virtue of focusing on 16S sequencing) but the mycobiome or virome could be equally important. A full metagenomic screening would be useful in understanding the roles of non-bacterial organisms in shaping host phenotype. A concern in this respect is that Figure 3D illustrates an upregulation of various immune pathways related to viral infection in TXwildling vs SPF lab mice. These results are potentially concerning when coupled with the lack of pathogen screening data – perhaps the TXwildling immune differences could be driven by the presence of a viral pathogen, with gut bacterial differences incidental? At the least, some discussion of the potential role of non-bacterial microbiome members is warranted. This also feels pertinent with respect to the gut mycobiome, which has been suggested to be a key player shaping immunological differences between lab and rewilded mice (Chen et al., 2023).

Finally, although adoption of a the TXwildling system is (albeit, at times somewhat indirectly) presented as something that could help with the reproducibility crisis, further research into the consistency of the TXwildling microbiome across different laboratory conditions and its phenotypic impacts would be needed to demonstrate this. The study doesn't demonstrate that the TXwildling microbiome has lower variability than the SPF microbiome when held in different facilities, only that the TXwildling microbiome is dominantly invasive against diverse SPF microbiota sourced from different facilities. Ultimately, it would have been most powerful to show not only that TXwildlings cause different SPF lab microbiomes to converge on a wildling microbiome (as in Fig. 1F) but that this then results in their phenotypic convergence (i.e. a more direct link to reproducibility). If this is beyond scope of this study, it would be useful at least to point to such future work to support the TXwildling approach, and perhaps frame this study more as a proof-of-concept for the development of such a model, rather than this specific wildling microbiome as being the standardised model that people could adopt.

2) Use of *Citrobacter rodentium* as the infection model

A gut bacterial infection system (*C. rodentium*) was used to assess how altered immune phenotypes of TXwildling mice might alter infection susceptibility. The choice of this bacterial infection seems potentially problematic, as differing degrees of microbial competition between Cr and either the TXwildling or SPF microbiota might instead be the root cause (Kamada et al. 2012). Unfortunately, it is not easy to disentangle the relative importance of host immunity from more direct bacterial resource competition in driving reduced susceptibility of TXwildlings to Cr. This could do with acknowledging and conclusions tempered accordingly. Given that Figure 3D illustrates an upregulation of various immune pathways pertinent to viral infection in TXwildling mice, one does wonder why a viral infection model was not chosen instead.

3) Statistical and numerical quantification of results and comparisons

We felt that some of the results presented could have further supported the conclusions with the inclusion of statistical analyses. Inclusion of stats in the PCA and PCoA plots within the publication would help quantify differences observed between SPF lab microbiomes, for example, which would strengthen the claim that TXwildling provides a standardized model if these significantly distinct microbiomes were standardized to a statistically indistinguishable composition. Moreover, stats comparing TXwildling to SPF microbiomes specifically can help quantify the differences between the two communities. Specifically, comparisons between alpha and beta diversities between groups could reinforce the claim that communities in the TXwildling gut more closely reflect those of the wildlings than lab SPF mice (which is visible in Fig. 1C & 1E). Statistics comparing the composition of gavaged versus co-housed mice would also back-up the conclusion that the TXwildling microbiome stably invades regardless of establishment method (line 129-133, 141-144).

On a similar note, it would also be useful to calculate and present comparisons of alpha- and beta-diversity indices of the different microbiomes included in this study. When comparing these metrics between the TXwildling and lab microbiomes, it can help readers understand the differences between a wildling versus lab microbiome, highlighting the weaknesses of lab microbiomes in simulating real-world microbiome scenarios (if they are more depauperate, lacking certain taxa etc). Meanwhile, such comparisons between the different SPF lab microbiomes can further highlight the disparity between laboratory sources, emphasizing the need for a standardized system.

Other points:

- In line 66-67 although it is stated “A substantial body of literature demonstrates that divergent microbiota among commercial vendors and research institutions is a primary reason for this global reproducibility crisis [11-25]”, many references in 11-25 merely point out the variation which exists in the SPF microbiome across facilities, but don’t provide evidence that this impacts reproducibility. We suggest separating out references here into those that support microbiome variation across facilities, and then those that can be cited to show or suggest (e.g. through evidence these microbiomes alter phenotypes) that such variation affects reproducibility.
- Line 77-8 – There are other studies beyond the authors’ lab that could be cited to reinforce this point
- Line 78-79 – State in what context/under what experimental conditions they are rapidly outcompetes.
- Line 88 – can you specify which are the ‘current candidates’ that are not complex/resilient enough?
- Line 91 – Is this referring to a standard microbiome? What specific criteria and minimum requisites should be used to select a “standard?” How should these be decided?
- Line 95 – Which characteristics of wild mice are pertinent to the issues of reproducibility and generalizability, etc.? Suggest mention examples
- Line 99 – Provide citations/evidence for members of the natural microbiota being highly adapted to the mouse gut
- Line 101-102 – what does being more resilient to ‘microbial challenges’ mean? Suggest clarify.
- Line 112 – Would be good to define what a “successful” transfer of a microbiome means/looks like early on.
- Line 122 – Useful to specify examples of the “key specialized infrastructure” needed
- Line 123 – Please define what a “TX system” is
- Line 125 – Clarify the protocol a bit more, was it that only one mouse was gavaged and then allowed to spread to the other four? Presumably this was to test whether TXwildling can spread in the face of numerical disadvantage, but clarify.
- Line 130 – Clarify if there was any statistical difference in microbiota composition between gavaged mice and those colonised passively by co-housing
- Line 144 – Clarify the grammar of “lab microbiota from all vendors,” sounds a bit like a mix of the different microbiomes with the current wording
- Line 187-188 – Worth reporting somewhere in manuscript what the key functional differences were between TXwildling and SPF mice
- Line 201 – Should provide examples of the “various aspects of host physiology” being discussed
- Figure 4 – Seems like TXwildling and wildling are still somewhat distinct based on the PCAs, but closer resemble each other than either do the SPF lab mice; stats would help here.
- Line 304-305 – Why were these T-cell chemoattractants chosen in particular?
- Line 373-380 – Could you specify whether these analyses were done after Citrobacter challenge and if so when, or in mice that weren’t challenged with Citrobacter at all? Its currently ambiguous from the positioning in the text and wording, so should be clarified.
- Line 378-380 – suggest make the specific pairwise comparisons clearer here e.g. “The adult human transcriptome was significantly enriched in the colon and lung transcriptome of TXwildlings and wildlings compared to lab mice, whereas lab mice showed a greater similarity to neonatal humans than TXwildlings or wildlings.”
- Line 390 – Day 15 seems like peak weight loss, at least in SPF group
- Line 399 – Use “facility” instead of “location” to avoid confusion with location in GI tract
- Line 400-402 – Back up the claim of variability in phenotypes being caused by microbiome variability across facilities with references
- Line 408 – What is a “TXsystem” exactly?
- Line 441 – Phenotype can be attributed to presence of, or absence of, the SFB?
- Line 454 – Reference [27] is maybe placed better with [51, 52] than at end of sentence
- Line 460 – What do key attributes I-V refer to? Refer back to intro
- Line 463 – How would researchers get the microbiome for the TXwildling system, and how would they use it? Perhaps this statement is a bit too bold/premature for this stage of the research?
- Line 682 – Just a thought, but it would be interesting to analyze the post-infection composition of the TXwildling

microbiome to see how it is changed by an infection challenge, especially in comparison to the SPF mice

- Line 700 – Typo “Golay” not “Golary” ?
- Fig. 6D – found colours of points in key are a bit too transparent to differentiate.

References Mentioned:

Chen, Y. H., Yeung, F., Lacey, K. A., Zaldana, K., Lin, J. D., Bee, G. C. W., ... & Cadwell, K. (2023). Rewilding of laboratory mice enhances granulopoiesis and immunity through intestinal fungal colonization. *Science immunology*, 8(84), eadd6910.

Kamada, Nobuhiko, et al. "Regulated virulence controls the ability of a pathogen to compete with the gut microbiota." *Science* 336.6086 (2012): 1325-1329.

Rosshart, S. P., Vassallo, B. G., Angeletti, D., Hutchinson, D. S., Morgan, A. P., Takeda, K., ... & Rehermann, B. (2017). Wild mouse gut microbiota promotes host fitness and improves disease resistance. *Cell*, 171(5), 1015-1028.

Reviewer #3

(Remarks to the Author)

Version 1:

Reviewer comments:

Reviewer #1

(Remarks to the Author)

- The housing conditions seem to be described twice now: L490-496 and L545-550
- Whilst L520 says "into only one out of 5 mice per cage", assuming the other mice were colonized via co-housing, L540/541 says that "each mouse was inoculated..."
- L564: 16S sequencing. Always use the term "16S rRNA gene amplicon sequencing" or at least "16S rRNA amplicon sequencing".
- "Shotgun metagenomic sequencing" is redundant. "Shotgun sequencing" is enough

Reviewer #2

(Remarks to the Author)

The authors have responded comprehensively to all of our initial concerns, including conducting follow-up evaluations of pathogen screening and metagenomic analysis of mycobionte elements, etc., within their samples. Additionally, they have provided good justification for their decisions to use a pathogen-containing wildling source of their microbiome, and why they opted to construct a novel system as opposed to testing the existing WildR system. Finally, additional contextualization of methods and results (particularly the addition of statistics to many areas we thought they were missing) has been helpful in communicating the results more thoroughly.

1. The rationale behind using wildlings and not WildR, and issues around pathogens have been comprehensively addressed, improving the paper's value and clarity. Our only remaining points on this are:

(i) it would be useful to clarify in the paper a little more clearly (as was done in responses) that a key 'hurdle' point for natural mouse microbiotas that the wildling model overcomes (compared to e.g. co-housing with pet store mice), is that it excludes HUMAN pathogens so can in theory be BSL1 and in normal mouse facilities. But that mouse pathogens intentionally remain, and you consider these an important component of the natural model.

[As a side note, one other hurdle likely to remain, is likely to be that many mouse facilities might still feel uneasy/not allow housing wildlings in the same building, as some of the mouse pathogens like *Spironucleus* are likely to be on whole-facility "Exclusion" lists, to avoid contamination risk to other precious or vulnerable mouse stocks.]

(ii) the metagenomic analyses of which non-bacterial microbes (fungi, viruses) were detected in the wildling and TXwildling don't seem to be fully presented, e.g. what other viruses and fungi were detected in the metagenomic screen that weren't targeted in the pathogen screening? This is currently only presented in summaries like "Fungi" or "Fungi-like protists" in Fig. 1e, but we can't see what's under the hood. Suggest include as this is important for anyone wishing to use the wildling microbiota in their studies. Perhaps a supplementary excel table tab?

2. "We have included statistical analyses in supplementary Figure S1".
Stats are better generally (thank you), tho a couple of points still:

- Please add to legend what statistical test was used in addition to the derived p-values.
- To strengthen/show the point that lab mouse microbiomes are less variable after TX treatment (e.g. Fig. S1g), key stat/figure would be to compare the average pairwise inter-facility (Taconic vs Jackson etc) microbiota distance and intra-facility distance before, and then after TX treatment. You would ideally show that (i) inter-facility before vs after comparison - the average inter-facility microbiome distance before TX treatment is much larger than after, and (ii) inter- vs. intra-facility distance comparison: the inter-facility distance far outstrips the intra-facility distance before TX treatment, but after TX treatment these are not statistically different (or far less so).

3. While we agree that the biological relevance of alpha diversity is uncertain, we think including the additional figure on richness and compositional barplots of the different microbiotas that was included in response to reviewers would be useful for readers – can you include it as a supplementary figure?

4. References: "This is evident in the divergent composition of their bacterial microbiome [30-35] and the underrepresentation of key non-bacterial constituents such as the virome and mycobiome [31,33, 36]." While the authors have added more refs for this point about natural microbiotas being different (30-35), these still seem to all or largely be using the wildling/WildR model in lab mice. What we meant is that there are quite a few papers that analyse other populations of gut microbiota in wild vs laboratory house mice, that could be cited here to broaden the point, since wildlings are from one site.

Examples would be:

- Wang, Jun, et al. "Dietary history contributes to enterotype-like clustering and functional metagenomic content in the intestinal microbiome of wild mice." *Proceedings of the National Academy of Sciences* 111.26 (2014): E2703-E2710.
- Kreisinger J, Cížková D, Vohánka J, Piálek J. Gastrointestinal microbiota of wild and inbred individuals of two house mouse subspecies assessed using high-throughput parallel pyrosequencing. *Mol Ecol.* 2014;23:5048–60.
- Bowerman KL, et al. Effects of laboratory domestication on the rodent gut Microbiome. *ISME Commun.* 2021;1:1–14.
- Hanski, Eveliina, et al. "Wild house mice have a more dynamic and aerotolerant gut microbiota than laboratory mice." *BMC microbiology* 25.1 (2025): 204.

5. The figure in the responses showing how gavaged and co-housed mice vary in the speed with which they diverge from lab mouse microbiome over time is useful... it encourages 3Rs (refinement) as shows gavage is not necessarily needed to get a good colonisation, it just speeds it up a little. Suggest accommodate this in Fig. S1.

6. Please clarify this in the legend of Fig. 6 also that the GSEA was performed on uninfected mice; with the infection challenge data the first panel in this figure, one could be forgiven for assuming the GSEA was also performed on the same mice.

„Laboratory mice engrafted with natural gut microbiota possess a wildling-like phenotype“

Reviewer #1

Remarks to the Author:

The manuscript was easy to read and presents convincing data on the relevance of the new mouse model presented. I have only one major comment and additional minor points.

Major Comment:

The author emphasises both the need for the community to have additional, more representative mouse models available and the need to provide standardised approaches for doing so. In contrast, I felt that the methods were not sufficiently described. If I understand one of the authors' aims correctly, namely to help the community by paving the way for the use of a model that is better than conventional mice and easier to use than wildlings, I would argue that it is not currently possible for readers to understand all the technical aspects behind the model and potentially reproduce the work. The methods around the preparation of the gut content stocks (e.g. sterile conditions? anaerobic conditions? volume of aliquots? etc.) and, perhaps more importantly, the gavage procedure (e.g. disposable aliquots, waiting time between melting and gavage, etc.). Diet information is also needed. Access to wildlings samples may also need to be clearer in the manuscript. These are only examples, and I urge the authors to think carefully and provide all the technical details necessary to reproduce this nice work.

We thank the reviewer for this important comment. For the preparation of the gut microbiota stocks, we have designed a protocol that could be performed in virtually every laboratory without specialized equipment. It was one of our goals to evaluate the robustness and resilience of natural microbiota under imperfect harvesting and processing conditions and to thereby probe, if the “TX system” would even be successful under suboptimal conditions. For this reason, we have not used an anaerobic chamber, since it is a specialized piece of equipment not readily available everywhere, instead we have processed our samples aerobically. Below you will find the detailed protocol in yellow that we have now added to the “Material and methods” section alongside additional detailed information regarding housing conditions etc (see lines 502 of the clean version).

More importantly, we would like to emphasize, that it is our primary goal to foster the field of natural microbiota-based mouse models globally. This manuscript is meant to encourage the field to colonize their mouse strain and model of interest with natural microbiota. Thus, glycerol stocks of natural microbiota are readily available from our biobank and we are happy to ship them out to laboratories worldwide.

Bacterial-gGut microbiome transplantation

"In general, the ileocecal microbial communities from terminal ileum and cecum of wildlings and SPF mice were transferred by a single oral gavage into only one out of 5 mice per cage.

To more effectively encompass the microbial diversity of the wildling colony, five wildlings were euthanized with CO₂ and subsequently harvested. The fur was disinfected with Bacillol, their abdominal cavity was opened under sterile conditions using sterile equipment and autoclaved surgical instruments in a biosafety cabinet class 2. Two separate sets of instruments were used, one set for the skin and one set for the intraabdominal preparation.

To minimize exposure to oxygen, the terminal ilea and ceca of all 5 donor mice were extracted from the peritoneal cavity without opening them and placed in a sterile bacteriological petri dish (Greiner Bio-One cat. # 633181). Subsequently, they were opened and manually extruded along the cephalocaudal axis to collect the fecal material into a sterile 100µm cell strainer (Corning Falcon cat. # 352360) placed inside of a sterile bacteriological petri dish. The collected fecal material was weighed and then diluted at a 1:2 ratio (g collected material : ml fluid) in sterile filtered cryoprotectant (PBS (Panbiotech cat. # P04-361000) containing 0.1% L-cysteine (Sigma-Aldrich cat. # C7352) and 20% glycerol (Sigma-Aldrich G2025)). The resulting suspension was subsequently mashed through the 100µm cell strainer to remove all insoluble particles, 1ml aliquots were prepared into 2mL externally threaded plastic screw-capped cryovials (Azenta Life Sciences cat. # BCS-2502), subjected to controlled freezing with 1°C per minute (Azenta Life Sciences CoolCell LX Cell cat. # BCS-405) and stored in a -80°C mechanical freezer. Prior to inoculation, the samples were thawed in a 37°C water bath and used for the inoculation within 10 minutes. Using a sterile syringe (B. Braun cat. # 9161708V) and gavage needle (Merck Cadence Science cat. # CAD9921-100EA), each mouse was inoculated one time with a total of 100µl of this suspension (equals 50mg of fecal material) via oral gavage. Harvest, preservation and transplantation of microbiota from SPF mice was performed in an identical fashion. After inoculation, all experimental mice were placed into clean cages followed by weekly cage changes.

Throughout the entire experiment, mice were housed under a 12:12 light:dark cycle (room temperature 20-24°C, humidity 45%-65%, air exchange rate of 15 times per hour) in a Greeline IVC system from Tecniplast inside of autoclaved microisolator cages (GM500) with autoclaved rodent chow (GRANOVIT AG, KLIBA NAFAG, 3437.PX.L15) and autoclaved tap water ad

libitum, 1x autoclaved sizzle pad 8g (ssniff), 1x autoclaved play tunnel (ssniff), 1x autoclaved smart home, and Aspen wood chip bedding (ssniff).”

Minor Comments:

1. The mice used in the experiments were relatively old (12-16 weeks) considering e.g. the development of immune responses; I think this deserves a comment / explanation, perhaps in the discussion.

We appreciate the reviewer’s insightful comment. We deliberately used older mice in our experiments, since we wanted to develop and validate a convenient and feasible transplantation system, a highly flexible platform for researchers around the world. This is why we deliberately tested, if the “TX system” would be capable of converting even adult mice (12 weeks and older) with a fully established microbiome and a mature immune system. This way, we would have established and validated an approach in which scientist would not depend on the technically more challenging inoculation of very young and small mice.

Of course, the reviewer is completely correct, it does make a lot of sense to inoculate young mice (e.g. 3-4 weeks of age) and this can also be done with the “TX system”.

We have emphasized this throughout the manuscript with the phrase “adult, fully colonized mice” (see lines 128, 134, 194, 412 of the clean version).

2. I found the parts about similarity to the human situation less relevant. While wildlings have been used in the past to successfully refute previous preclinical data obtained using standard laboratory mice, the main point is to enrich the range of models available. A devil's advocate could probably list several negative aspects of TXwildlings. The authors may wish to provide a more balanced view.

We thank the reviewer for this thoughtful comment. We agree, TX wildlings do enrich the range of models available and this certainly is a main point of the manuscript.

Nevertheless, to the authors, another main point and reason to use natural microbiota-based mouse models is that they more closely resemble human physiology compared to SPF mice. These models feature a more mature immune system, which is essential for understanding complex immune responses seen in humans. The greater similarity TXwildlings and wildlings share with humans is a central reason why we consider them a valuable tool and ultimately model for translational biomedical research. Of course, wildlings are the more complete and also versatile model and we acknowledge that TX wildlings do come with limitations which we state as follows in line 431 of the clean version:

“However, it is important to emphasize that the natural microbiota of TXwildlings are limited to the gut, while wildlings are colonized with wild mouse microbiota across all epithelial barrier sites, including the skin, vagina, and lung [31]. Additionally, TXwildlings are not exposed to natural microbiota throughout all developmental stages, potentially overlooking effects during ontogeny [31, 34].”

However, TXwildlings do have important advantages over other natural microbiota-based mouse models as explained in line 463 of the clean version.

“We developed and validated the TXwildling system – a scalable, standardizable, and controlled approach to make natural microbiota-based models globally accessible on various genetic backgrounds.”

3. L68: “the metagenome, the combination of the microbiome and the host genome”; this is incorrect; <https://doi.org/10.1186/s40168-020-00875-0>; <https://doi.org/10.1186/s40168-015-0094-5>

We are thankful for this mindful commentary and would like to inform you that we took the definition from Stappenbeck and Virgin as they state: “The metagenome is defined as the total host genome and the associated microbiome genome” [1]. However, we acknowledge the concern that this might be misleading and have changed the respective sentence (line 67 of the clean version).

“It is crucial to recognize that the ~~metagenome, the~~ combination of the microbiome and the host genome ~~drives~~ the mammalian phenotype [17].”

4. L74: naturally coevolved microbes, not microbiota; the term microbes may be more appropriate also in other parts of the manuscript?

We thank the reviewer for this suggestion. We changed the term microbiota to microbes in the respective sentence (see line 72 of the clean version).

“Their microbiota are the result of repeated germ-free rederivation and recolonization in restrictive laboratory environments leading to a complete loss of naturally co-evolved ~~microbiota~~ microbes.”

5. The terms "bacterial microbiome" (L76) or "bacterial gut microbiome" (L675), perhaps also elsewhere in the manuscript, are inaccurate.

We are thankful for this advice and changed it to "microbiome" in lines 76 and 502 of the clean version.

"This is evident in the divergent composition of their ~~bacterial~~ microbiome [30-35] and the underrepresentation of key non-bacterial constituents such as the virome and mycobiome [31, 33, 36]."

~~"Bacterial gGut microbiome transplantation"~~

6. L211: While there were (always past tense in the methods and results)

We thank the reviewer for carefully reading the manuscript and bringing up this point. We have changed "are" to "were" in line 229 of the clean version.

"While there ~~are were~~ 1422 genes significantly differently regulated between TXwildlings and lab mice, only two genes significantly differed between TXwildlings and wildlings."

Reviewer #2 and #3

Remarks to the Author:

In this study Runge et al. demonstrate some interesting characteristics of a wildling mouse microbiome transplanted into laboratory mice (referred to as TXwildling mice in the paper). The authors show that the gut microbiome of wildling mice consistently invades and apparently replaces the microbiota of co-housed SPF cagemates, regardless of the ratio between TXwildling and SPF mice. The paper also comprehensively characterises immune phenotypes of TXwildling mice both locally in the gut but also at distant body sites such as liver and lung, showing them to be rather similar to those of wildlings from which their microbiome is derived, and also more comparable to a mature human than that of SPF lab mice. Moreover, both the TXwildling and wildling transcriptomes (colon and lung) resembled that of adult human more closely than did that of lab mice. These results suggest that compared to the typical microbiome of SPF lab mice, the TXwildling microbiome may provide a more stable platform which better recapitulates the phenotypes of wild mice and adult humans, for use in biomedical research. The authors argue that since the creation of natural microbiota-based mouse models come with significant logistical hurdles, this TXwildling system presents a feasible alternative to convert standard lab mice to natural microbiota-based models.

We see significant value in this work, and agree that such a natural mouse model with a resilient wild-derived microbiome holds real value for mouse research. However, we do have some concerns with the work as presented that we think need addressing to maximise its impact.

Major comments:

1) Value of TXwildlings as a standardized wild mouse microbiome model

Throughout the text the authors highlight that the TXwildling system provides a stable wildling-sourced microbiome which they anticipate could provide a useful tool for standardizing lab mouse microbiota in future studies. However, in light of the study's scope and the data presented, currently this seems a rather strong claim, for reasons expanded upon below.

The authors state (lines 121-122) that natural microbiota-based models demand significant funding, expertise and specialised infrastructure. But that widespread adoption of TXwildlings could make them more globally accessible (line 463). In order to assess how TXwildlings

alleviate these hurdles, it would be useful to be more explicit what exactly these challenges are and how TXwildlings overcomes them. Isn't one of the major challenges of natural microbiota models the potential presence of pathogens, and thus the need to house them outside standard lab animal facilities? If so, it is unclear how TXwildlings would overcome this at present, as no pathogen screening results have been presented (and it is not clear in the original wildling paper what might transmit via gut contents). We would therefore suggest including comprehensive pathogen screening data (and ideally metagenomic data to assess viruses that may not have dedicated assays) to support that they are pathogen-free, or otherwise modify this claim.

We thank the reviewers for their valuable feedback and suggestions. We appreciate the opportunity to clarify the challenges associated with natural microbiota-based mouse models and how TXwildlings may offer solutions to these hurdles.

1. Expensive/specialized infrastructure:

- *Natural microbiota-based mouse models usually require specialized infrastructure, e.g. outdoor enclosures for the “re-wildling approach” [2], or BLS2 and even BSL3 animal facilities for “the dirty mouse pet store co-housing approach” [3]. This is very different for wildlings and we would like to clarify common misconceptions:*
 - *Once established, wildlings and thus also TXwildlings do not require expensive and specialized infrastructure such as outdoor enclosures. The animals can be housed in a regular animal facility, in conventional IVCs [4-6] and even under static housing conditions [7].*
 - *Most importantly, wildlings and TXwildlings do carry mouse pathogens, they are not meant to be free of murine pathogens, since these pathogens are essential for the functionality of the model system (please also see our response to the next question regarding WildR mice). However, wildlings and TXwildlings are securely devoid of human pathogens such as LCMV and Hantavirus. Thus, wildlings and also TXwildlings do not require BSL2 or BSL3 conditions, they are BSL1. Based on our long-term experience with these model systems, we have developed SOPs and implemented them in several animal facilities [8] illustrating that wildlings can be safely housed within the same facility with conventional SPF mice under BSL1 conditions without compromising safety or contamination control. We would like to emphasize, that the required permissions were even granted under the extraordinarily strict German rules and regulations. While a separate room might still be beneficial in some cases, it is not strictly necessary.*

Thus, TXwildlings do not require specialized and expensive infrastructure. The required pathogen testing has now been performed and added to the manuscript as Table S1 (please also see our answer to the next question).

2. Technical challenges:

- Some natural microbiota-based models, like wildlings, require catching wild mice [7, 9], sequential pathogen testing to exclude human pathogens and performing embryo transfer or fostering [5-7], which is technically demanding and resource-intensive. TXwildlings circumvent this issue as our model relies on oral gavage of material sourced from an already existing and well-established BSL1 wildling colony. More importantly, we would like to emphasize, that it is our primary goal to foster the field of natural microbiota-based mouse models globally. Thus, glycerol stocks of natural microbiota are readily available from our biobank and we are happy to ship them out to laboratories worldwide. Thus, TXwildlings are highly accessible, scalable and easier to implement.*

3. Time consuming:

- The time needed for establishing other microbiota-based mouse models by co-housing, breeding or sequential infection can be lengthy, potentially making experiments time-consuming [10]. TXwildlings offer a significant advantage in this regard. They allow for faster microbiome establishment and experimental timelines, ensuring that experiments can be completed within a more efficient timeframe.*

In line with this, we are unsure why the authors chose to use wildling microbiome as the source microbiome for this study and to recommend (line 112) as opposed to the WildR microbiome previously generated by this research group (Rosshart et al., 2017), which has already been pathogen-screened and is now a commercially-available 'SPF' wild mouse microbiome model. This would seem to be more readily 'safe' and adoptable by others keen to naturalise their mouse models' microbiota through co-housing.

We are grateful for this question, especially your important interest in pathogen screening. We have now performed a very broad and comprehensive characterization of pathogens present in TXwildlings compared to wildlings and conventional SPF mice. To create the most reliable dataset, we have decided to use two independent, validated and commercial testing strategies, PCR and serology. We now provide these data in Table S1. This table clearly illustrates that wildlings and TXwildlings carry an identical profile of murine pathogens, but that they are devoid of human pathogens and match BSL1 requirements.

Please also allow us to explain in greater detail, why wild mouse microbiome-reconstituted (WildR) mice are a valuable model system, but not comparable to wildlings or TXwildlings. We have established WildR mice in 2017 [9]. They are inbred laboratory C57BL/6 mice colonized with natural microbiota that were sourced from wild mice while excluding specific pathogens. WildR mice exhibited reduced inflammation and increased survival following influenza virus infection and improved resistance against mutagen/inflammation-induced colorectal tumorigenesis. Since WildR mice are free of pathogens, the underlying mechanism of protection must be mediated by natural microbiota and not pathogens. We have called this phenomenon “host fitness-promoting traits of natural microbiota”. However, WildR mice are missing the impact of pathogens on host physiology crucial for modelling humans and free-living animals [3, 7, 11]. For this reason, we have developed wildlings combining “host fitness-promoting traits of natural microbiota” with „pathogenic experience“, to create a mouse model with a fully mature immune system to better model human immunity for basic as well as preclinical biomedical research questions [7].

This is why we deliberately chose pathogen-containing wildling microbiome as the source microbiome for this study. We wanted to transfer pathogens, because of the importance of pathogen exposure in immune system maturation and thus translational research value as well as applicability to real-world scenarios. WildR mice represent a pathogen-free wild mouse microbiome model that allows for the discovery of microbiota-mediated mechanisms, that cannot be found in conventional SPF mice. However, as explained above WildR mice are missing the important immune maturation that corresponds with pathogenic exposure and that is essential for immune system development as well as maturation (e.g. development of tissue-resident memory cells, etc.) and an enhanced translational research value.

The manuscript largely implies the bacterial microbiota as being the key factor driving different immune phenotypes (largely implicitly, by virtue of focusing on 16S sequencing) but the mycobiome or virome could be equally important. A full metagenomic screening would be useful in understanding the roles of non-bacterial organisms in shaping host phenotype.

A concern in this respect is that Figure 3D illustrates an upregulation of various immune pathways related to viral infection in TXwildling vs SPF lab mice. These results are potentially concerning when coupled with the lack of pathogen screening data – perhaps the TXwildling immune differences could be driven by the presence of a viral pathogen, with gut bacterial differences incidental? At the least, some discussion of the potential role of non-bacterial microbiome members is warranted. This also feels pertinent with respect to the gut mycobiome, which has been suggested to be a key player shaping immunological differences between lab and rewilded mice (Chen et al., 2023).

We thank the reviewers for this pivotal comment, it was not our intention to promote bacterial microbiota as the key factor. We fully agree that non-bacterial components of the gut microbiome play an important role, this is why we have characterized the mycobiome and virome in earlier studies [4, 7]. For this reason, we have now performed shotgun metagenomics analysis of lab mice, TXwildlings and wildlings alongside ITS sequencing and a broad pathogen characterization (see above and new Table S1). We found, significant quantitative and qualitative differences among the groups:

*TXwildlings as well as wildlings carry significantly more non-bacterial microorganisms, like protozoan, protists, fungi, and viruses than conventional lab mice. Especially fungi in conventional lab mice were of a very low biomass, while TXwildlings and wildlings displayed a substantial fungal biomass dominated by *Kazachstania pintolopesii*. It is important to note that the overall composition between wildlings and TXwildlings was very similar suggesting a complete engraftment of not only bacteria, but also non-bacterial constituents of the microbiome. Importantly, we were successful in transferring pathogens from wildlings to TXwildlings. We show these new and very insightful data in Table S1 and Figure S1f-g and have adjusted the manuscript accordingly from line 142 of the clean version:*

*“There is an increasing appreciation that non-bacterial constituents of the microbiome such as fungi and viruses play an important physiological role [12-17]. Therefore, we assessed whether non-bacterial components were also successfully transferred to TXwildlings. Using shotgun metagenomics and internal transcribes space (ITS) sequencing, TXwildlings as well as wildlings were found to carry significantly more non-bacterial microorganisms, like protozoan, protists, fungi, and viruses than conventional lab mice (**Figure S1e**). In conventional lab mice the fungal biomass was very low, while TXwildlings and wildlings displayed a substantially higher fungal biomass dominated by *Kazachstania pintolopesii* (**Figure S1f**). Further, pathogenic experience is a wildling-defining characteristic and crucial for the functionality of several natural microbiota-based mouse models [31, 47, 48]. Hence, we characterized the pathogen profile using serology and PCR and found that we were successful in transferring pathogens from wildlings to TXwildlings (**Table S1**). This suggests a complete engraftment of not only bacteria, but also non-bacterial constituents of the microbiome alongside pathogens.”*

Finally, although adoption of a the TXwildling system is (albeit, at times somewhat indirectly) presented as something that could help with the reproducibility crisis, further research into the consistency of the TXwildling microbiome across different laboratory conditions and its phenotypic impacts would be needed to demonstrate this. The study doesn't demonstrate that the TXwildling microbiome has lower variability than the SPF microbiome when held in different facilities, only that the TXwildling microbiome is dominantly invasive against diverse SPF microbiota sourced from different facilities. Ultimately, it would have been most powerful to

show not only that TXwildlings cause different SPF lab microbiomes to converge on a wildling microbiome (as in Fig. 1F) but that this then results in their phenotypic convergence (i.e. a more direct link to reproducibility). If this is beyond scope of this study, it would be useful at least to point to such future work to support the TXwildling approach, and perhaps frame this study more as a proof-of-concept for the development of such a model, rather than this specific wildling microbiome as being the standardised model that people could adopt.

Yes, we agree with the reviewer that we are working with strong, but “somewhat indirect” data regarding this particular point. Further, we are thankful that the reviewer acknowledges that experiments in different facilities (basically a multicenter trial) would be beyond the scope of our current study. Thus, we followed the reviewers advice, adjusted our corresponding language and “frame this study as a proof-of concept study” that needs further validation. Importantly, this was done in the final conclusion of the discussion in line 459 of the clean version:

“Taken together, microbiota standardization addresses the reproducibility crisis by resolving the issue of divergent microbiota across institutions, though no suitable candidate has been identified to date [37][18]. In this work, we presented an evidence-based rationale and proof-of-concept study to use natural gut microbiota as candidate for standardization due to their key attributes mentioned in the introduction (I-V). We developed and validated the TXwildling system – a scalable, standardizable, and controlled approach to make natural microbiota-based models globally accessible on various genetic backgrounds. Although this need to be further validated and tested in different locations, we anticipate that the widespread adoption of TXwildlings will may enable the discovery of novel treatments, enhance reproducibility, reduce research costs, and improve the safety and success of translational efforts, ultimately advancing human health.”

2) Use of *Citrobacter rodentium* as the infection model

A gut bacterial infection system (*C. rodentium*) was used to assess how altered immune phenotypes of TXwildling mice might alter infection susceptibility. The choice of this bacterial infection seems potentially problematic, as differing degrees of microbial competition between Cr and either the TXwildling or SPF microbiota might instead be the root cause (Kamada et al. 2012). Unfortunately, it is not easy to disentangle the relative importance of host immunity from more direct bacterial resource competition in driving reduced susceptibility of TXwildlings to Cr. This could do with acknowledging and conclusions tempered accordingly. Given that Figure 3D illustrates an upregulation of various immune pathways pertinent to viral infection in TXwildling mice, one does wonder why a viral infection model was not chosen instead.

We agree with the reviewer and acknowledge the comprehensive understanding of the C. rodentium infection model. Our steady-state multi-organ systems analysis examining immunological barrier sites, central non-lymphoid organs, lymphoid organs, and blood clearly demonstrated that TXwildlings and wildlings share key immunological characteristics while being significantly different from genetically identical conventional SPF mice, implying that they may respond similarly to actual physiological/pathophysiological challenges. To confirm this conclusion, we challenged and assessed the experimental groups to evaluate if the fecal transplantation was successful in changing the actual phenotype of lab mice to that of wildlings. To achieve this, we wanted to utilize a robust, very well described and broadly used model system. Further, we wanted to acknowledge the mammalian metaorganism and therefore utilize a challenge involving both host and microbiota factors in its pathophysiology. C. rodentium was the model system fulfilling these criteria in the most satisfying fashion. Of course, we do agree that the complexity of the C. rodentium model makes it very difficult to entangle the relative contribution of the host and microbiota. But while it makes a mechanistic insight difficult, we argue that specifically because of C. rodentium's broader impact, it is well suited to our intentions.

3) Statistical and numerical quantification of results and comparisons

We felt that some of the results presented could have further supported the conclusions with the inclusion of statistical analyses. Inclusion of stats in the PCA and PCoA plots within the publication would help quantify differences observed between SPF lab microbiomes, for example, which would strengthen the claim that TXwildling provides a standardized model if these significantly distinct microbiomes were standardized to a statistically indistinguishable composition. Moreover, stats comparing TXwildling to SPF microbiomes specifically can help quantify the differences between the two communities. Specifically, comparisons between alpha and beta diversities between groups could reinforce the claim that communities in the TXwildling gut more closely reflect those of the wildlings than lab SPF mice (which is visible in Fig. 1C & 1E). Statistics comparing the composition of gavaged versus co-housed mice would also back-up the conclusion that the TXwildling microbiome stably invades regardless of establishment method (line 129-133, 141-144).

We agree that a statistical description of the ordination plots emphasizes the change in the lab mouse microbiota after fecal transplantation from wildlings, thank you for this important comment. We have included statistical analyses in supplementary Figure S1. To compare the change in microbiota composition after transplantation (that is either receiving wildling feces or lab feces in the respective experiments) we opted to show the difference in each recipient

group and their baseline counterparts to fecal donor. Thus, our goal is to demonstrate the shift after exposing an already fully established microbiome to another microbiome. Essentially, we are demonstrating the success by which an “invading” microbiome can replace the existing microbiome. To prevent an artificial inflation of distances used in such a comparison, we decided to compare to the centroid of the donor microbiome rather than comparing each individual point to each individual point.

On a similar note, it would also be useful to calculate and present comparisons of alpha- and beta-diversity indices of the different microbiomes included in this study. When comparing these metrics between the TXwildling and lab microbiomes, it can help readers understand the differences between a wildling versus lab microbiome, highlighting the weaknesses of lab microbiomes in simulating real-world microbiome scenarios (if they are more depauperate, lacking certain taxa etc). Meanwhile, such comparisons between the different SPF lab microbiomes can further highlight the disparity between laboratory sources, emphasizing the need for a standardized system.

Point-by-point-response Figure 1.

Lab mice were purchased from different vendors and transplanted with wildling fecal material. (a) Alpha diversity measures (Chao1, Fisher, Shannon) were calculated for lab microbiota of the different vendors before and 28 days post-transplantation with wildling fecal microbiota and compared to alpha diversity of wildlings. (b) Relative abundances at the family level were calculated of the lab microbiota of the different vendors before and after transplantation as well as for wildlings. Data are from one (Charles River, Envigo, Janvier, Jackson), three (Taconic) or four (wildlings) independent experiments.

Thank you for raising this point. The Figure above additionally illustrates that laboratory mouse microbiota are very different from each other and that the microbiota of TXwildlings are more

comparable. Due to the unique nature of the wildling microbiome and its high complexity, alpha diversity indices are still difficult to understand with regards to their biological impact. This is why we have decided to not show these results in the manuscript. Still, the alpha diversity and the change in the different vendors' diversity after transplantation clearly demonstrates the success in the transplantation of natural microbiota of wildlings into the commercial vendors.

Minor Comments:

1. In line 66-67 although it is stated “A substantial body of literature demonstrates that divergent microbiota among commercial vendors and research institutions is a primary reason for this global reproducibility crisis [11-25]”, many references in 11-25 merely point out the variation which exists in the SPF microbiome across facilities, but don't provide evidence that this impacts reproducibility. We suggest separating out references here into those that support microbiome variation across facilities, and then those that can be cited to show or suggest (e.g. through evidence these microbiomes alter phenotypes) that such variation affects reproducibility.

We thank the reviewer for this suggestion and separated out references accordingly (see line 69 of the clean version).

2. Line 77-8 – There are other studies beyond the authors' lab that could be cited to reinforce this point

We are thankful for this suggestion and have added additional references that support our statement (see line 76 of the clean version).

“This is evident in the divergent composition of their ~~bacterial~~-microbiome [30-35] and the underrepresentation of key non-bacterial constituents such as the virome and mycobiome [31, 33, 36].”

3. Line 78-79 – State in what context/under what experimental conditions they are rapidly outcompetes.

We thank the reviewer for this suggestion. We have added the experimental condition to the text of the manuscript and added one additional reference (see line 78 of the clean version).

“In addition, they are rapidly outcompeted and replaced by natural microbiota in a co-housing setup [31, 34] and undergo significant changes in community structure upon even minor environmental perturbations, such as mouse husbandry conditions [12] and transportation [14,

22], resulting in divergent and location-specific conventional lab microbiota and phenotypes across different institutions [37].”

4. Line 88 – can you specify which are the ‘current candidates’ that are not complex/resilient enough?

We thank the reviewer for pointing out this inaccuracy. We have revised the respective paragraph and have added additional references and explanations in line 87 of the clean version.

“~~Thus-Hence~~, a retrospective analysis will rarely clarify the differences observed [19, 43-45]. ~~However, current candidates,~~ Therefore, conventional SPF microbiota, cannot be used for standardization since they lack the required complexity and resilience, resulting in the re-emergence of divergent microbiota and irreproducible data across different institutions. Further, defined microbial consortia can solve the problem of divergent microbiota across different institutions. However, these artificial consortia remain stable only under gnotobiotic housing conditions and lack the complexity of a naturally co-evolved microbiota. As a result, these approaches prioritize reproducibility at the expense of physiological relevance. Therefore, ~~the~~ goal must be to select a standard with the necessary complexity and resilience to overcome narrow, non-generalizable local phenotypes and create robust mouse models across different institutions [37].”

5. Line 91 – Is this referring to a standard microbiome? What specific criteria and minimum requisites should be used to select a “standard?” How should these? be decided?

Line 95 – Which characteristics of wild mice are pertinent to the issues of reproducibility and generalizability, etc.? Suggest mention examples

We are grateful for this suggestion. We present an evidence-based rationale and listed the criteria and minimum requisites to select a standard microbiome later in the text in line 99 of the clean version. Natural microbiota of wild mice fulfill these criteria, this is why we suggest to use microbiota with such characteristics as a standard.

“We suggest that naturally co-evolved microbiota of wild mice possess the necessary biological complexity alongside unique characteristics to serve as a suitable candidate for successful standardization [37]: I) They can be harvested, viably preserved, frozen, bio-banked, thawed, transferred, and engrafted into lab mice [30]. II) Natural microbiota are well characterized in their overall composition [31] and have evolved under evolutionary pressure in a challenging environment. As a result, they are highly adapted to the mouse gut, III) remain stable in the multi-generational offspring of lab mice [30, 31], IV) outcompete lab microbiota and possess

remarkable resilience against strong environmental disturbances like microbial challenges [31], antibiotics [31], change of diet [31, 34, 35], and a stable core microbiome upon shipment and across multiple institutions with different husbandry conditions [30, 31, 33, 35, 46]. V) Finally, natural microbiota-based mouse models, such as wildlings, have a superior translational research value over lab mice since their mature immune system better mirrors human physiology [31, 46-49]. However, creating natural microbiota-based models demands significant funding, expertise, and specialized infrastructure, making them inaccessible for widespread use [30, 31, 40, 47, 48, 50].“

7. Line 99 – Provide citations/evidence for members of the natural microbiota being highly adapted to the mouse gut

We thank the reviewer for this question. Please let us clarify this point. There is no data on specific microorganisms present within natural microbiota, instead natural microbiota as a whole community have been clearly shown to be better adapted to the gastrointestinal tract of mice than conventional SPF microbiota [7]. In general, microbiota have co-evolved with their respective hosts over millions of years, making them species-dependent [19]. Hence, conventional lab SPF microbiota can outcompete human, termite, fish and soil microbiota, showing that they are better adapted to the mouse gastrointestinal tract [20]. Natural microbiota are so well-adapted to the gut niche, that they can even outcompete such conventional lab SPF microbiota with striking speed in a co-housing setup, showing their superior adaptation as a community [7].

8. Line 101-102 – what does being more resilient to ‘microbial challenges’ mean? Suggest clarify.

The “microbial challenge” was the ingestion of fecal pellets, the experimental setup was co-housing. Wildlings with natural microbiota were co-housed with genetically identical mice with conventional lab SPF microbiota. In this scenario, wildlings did not display any disturbances of their microbiota, their microbiome remained unchanged despite ingesting fecal pellets from conventional lab SPF mice [7].

9. Line 112 – Would be good to define what a “successful” transfer of a microbiome means/looks like early on.

We thank the reviewer for raising this point.

A successful microbiota transfer ultimately means a successful conversion of a conventional SPF mouse into a natural microbiota-based model system. We evaluated this success by acknowledging the mammalian metaorganism and assessing structural as well as functional aspects of the microbiome and the host as broad as possible:

1. Microbiome:

- *Structure:*
 - *Conversion of community structure assessed by 16S rRNA gene profiling, shotgun metagenomics, ITS sequencing and pathogen characterization via two independent methods (Figure 1, Figure S1 as well as Table S1)*
- *Function:*
 - *Conversion of the function of the microbiome by utilizing metabolomics as an accepted surrogate parameter of function (Figure 2, corresponding supplementary Figure S2)*

2. Host:

- *Structure including functional features/aspects (e.g., antibodies etc.):*
 - *Phenotypic conversion of the host through a steady-state multi-organ systems analysis characterizing representative organs local and distant to the gut at multiple levels utilizing multiple independent readouts (Figure 3-5, corresponding supplementary Figures S3-S5)*
- *Function:*
 - *Conversion of actual phenotype in the context of a pathophysiological challenge (Figure 6a and 6b) and better representation of human immunity (Figure 6c and 6d)*

10. Line 122 – Useful to specify examples of the “key specialized infrastructure” needed

We appreciate the reviewer’s suggestion and have added examples of key specialized infrastructure to the text (see line 123 of the clean version).

Since the creation of natural microbiota-based mouse models demands significant funding, expertise, and specialized infrastructure like BSL3 laboratories, outdoor enclosures or specialized indoor enclosures. To circumvent this, our study aimed to develop and validate a feasible TX system to convert lab mice into natural microbiota-based models.

11. Line 123 – Please define what a “TX system” is

We thank the reviewer for this comment. The “TX system” is defined earlier in the manuscript in line 114 of the clean version as a “controlled natural gut microbiota transplantation ~~(TX)~~ system (TX system)”.

12. Line 125 – Clarify the protocol a bit more, was it that only one mouse was gavaged and then allowed to spread to the other four? Presumably this was to test whether TXwildling can spread in the face of numerical disadvantage, but clarify.

We appreciate the reviewer’s request for clarification. In all experiments, a single mouse was gavaged and transmission was allowed to occur naturally to the other four cage mates. This approach was chosen to maintain a simple and accessible protocol, facilitating easy replication of the method. Since this was also a major comment by reviewer 1, we have now added the following detailed paragraph to the “Material and methods” section in line 502 of the clean version:

“In general, the ileocecal microbial communities from terminal ileum and cecum of wildlings and SPF mice were transferred by a single oral gavage into only one out of 5 mice per cage. To more effectively encompass the microbial diversity of the wildling colony, five wildlings were euthanized with CO₂ and subsequently harvested. The fur was disinfected with Bacillol, their abdominal cavity was opened under sterile conditions using sterile equipment and autoclaved surgical instruments in a biosafety cabinet class 2. Two separate sets of instruments were used, one set for the skin and one set for the intraabdominal preparation.

To minimize exposure to oxygen, the terminal ilea and ceca of all 5 donor mice were extracted from the peritoneal cavity without opening them and placed in a sterile bacteriological petri dish (Greiner Bio-One cat. # 633181). Subsequently, they were opened and manually extruded along the cephalocaudal axis to collect the fecal material into a sterile 100µm cell strainer (Corning Falcon cat. # 352360) placed inside of a sterile bacteriological petri dish. The collected fecal material was weighed and then diluted at a 1:2 ratio (g collected material : ml fluid) in sterile filtered cryoprotectant (PBS (Panbiotech cat. # P04-361000) containing 0.1% L-cysteine (Sigma-Aldrich cat. # C7352) and 20% glycerol (Sigma-Aldrich G2025)). The resulting suspension was subsequently mashed through the 100µm cell strainer to remove all insoluble particles, 1ml aliquots were prepared into 2mL externally threaded plastic screw-capped cryovials (Azenta Life Sciences cat. # BCS-2502), subjected to controlled freezing with 1°C per minute (Azenta Life Sciences CoolCell LX Cell cat. # BCS-405) and stored in a -80°C mechanical freezer. Prior to inoculation, the samples were thawed in a 37°C water bath and used for the inoculation within 10 minutes. Using a sterile syringe (B. Braun cat. # 9161708V)

and gavage needle (Merck Cadence Science cat. # CAD9921-100EA), each mouse was inoculated one time with a total of 100µl of this suspension (equals 50mg of fecal material) via oral gavage. Harvest, preservation and transplantation of microbiota from SPF mice was performed in an identical fashion. After inoculation, all experimental mice were placed into clean cages followed by weekly cage changes.

Throughout the entire experiment, mice were housed under a 12:12 light:dark cycle (room temperature 20-24°C, humidity 45%-65%, air exchange rate of 15 times per hour) in a Greeline IVC system from Tecniplast inside of autoclaved microisolator cages (GM500) with autoclaved rodent chow (GRANOVIT AG, KLIBA NAFAG, 3437.PX.L15) and autoclaved tap water ad libitum, 1x autoclaved sizzle pad 8g (ssniff), 1x autoclaved play tunnel (ssniff), 1x autoclaved smart home, and Aspen wood chip bedding (ssniff).”

13. Line 130 –Clarify if there was any statistical difference in microbiota composition between gavaged mice and those colonised passively by co-housing

We thank the reviewer for this excellent question. We have added a PCA that highlights the mice receiving transplantation to Figure S1. We also analyzed the distance of each sample to the centroid of the lab mice before transplantation. This comparison reveals a statistically significant difference in distance between the mice receiving gavage and the co-housed mice. However, this difference is gone from day 10 post-transplantation onward. In simple words, the colonization outcome was identical, the data just show that the gavaged mice were colonized first.

Point-by-point-response Figure 2.

Fecal material from wildlings was transplanted by oral gavage into a lab mouse, which was housed with four lab mice in the same cage. (a) PCoA based on the Jaccard distance, comparing the microbiota of lab mice before and at the indicated days after transplantation with wildling fecal material to those of the wildling donor mice. Those mice receiving transplantation are highlighted. (b) The distance of each mouse at each timepoint to the centroid of the lab mice is calculated. The graph compared mice that received transplantation to those that were co-housed. Data are from three independent experiments with one recipient mouse co-housed together with four cage mates in each experiment. Panel (a) is also shown in Figure S1c.

14. Line 144 – Clarify the grammar of “lab microbiota from all vendors,” sounds a bit like a mix of the different microbiomes with the current wording

We thank the reviewer for this suggestion. We used a clearer sentence structure now (see line 162 of the clean version).

*“Conversely, the exposure to lab microbiota, **regardless of the vendor they originate from, from all vendors,** did not affect the composition of the natural microbiota of wildlings.”*

15. Line 187-188 – Worth reporting somewhere in manuscript what the key functional differences were between TXwildling and SPF mice

We appreciate the reviewer’s question. Our primary question is whether the functional capability of the microbiome in TXwildlings is comparable to that of wildlings, to assess, if the engraftment was successful. Identifying the detailed key functional differences between TXwildlings and SPF mice was not the question or within the scope of our study and needs further research. Of course, all of our data will be deposited and are available for the community for further in-depth analyses.

16. Line 201 – Should provide examples of the “various aspects of host physiology” being discussed

We are thankful for the reviewer’s suggestion. To clarify, we have specified a key aspect of host physiology that differs between wildlings and laboratory mice (see line 217 of the clean version).

*However, wildlings and laboratory mice differ significantly in various aspects of host physiology, **especially features of the immune system** [31].*

17. Figure 4 – Seems like TXwildling and wildling are still somewhat distinct based on the PCAs, but closer resemble each other than either do the SPF lab mice; stats would help here.

We thank the reviewer for this suggestion. Indeed, we have added statistical analyses for the PCAs presented in Figures 3 and 4 to the corresponding Supplementary Figure (Figures S3 and S4). For example, for the PCA of the colon RNA sequencing data, we compared the distance of the lab and TXwildling samples to the centroid of the wildlings. The statistical analysis demonstrates that TXwildlings are significantly closer to the wildlings than the lab mice. In contrast, the distance between TXlab mice and lab mice as well as wildlings and lab mice does not change significantly.

Point-by-point-response Figure 3

RNA sequencing was performed on colon tissue from lab mice, TXwildlings, wildlings and TX lab mice. (a) PCA of samples from lab mice, TXwildlings and wildlings. (b) Distance measure to the centroid of the wildlings is compared between lab mice and TXwildlings. (c) PCA of samples from lab mice, wildlings and TXlab mice. (d) Distance measure to the centroid of the lab mice is compared between wildlings and TXlab mice. Significances were tested using unpaired Mann-Whitney U tests. ** $p < 0.01$. Panel (a) is part of the manuscript in Figure 3b. Panels (b-d) are part of Figure S3b-d.

18. Line 304-305 – Why were these T-cell chemoattractants chosen in particular?

CCL5 and CXCL10 are both chemokines that play a key role in the recruitment of immune cells, including T cells. This is particularly relevant given the observation of increased numbers of CD4⁺ and CD8⁺ T cells in the liver. Both chemokines are known to attract T cells, particularly CD8⁺ T cells and T helper cells. In addition to these T cell chemoattractants, there are other chemoattractants that may act as T cell attractants in some situations. Their concentration in the liver shows the same tendency as shown in the manuscript.

Point-by-point-response Figure 4

The concentrations of different chemoattractants were measured in liver homogenates of lab mice, TXwildlings and wildlings using an MSD multiplex assay. Significances were tested using unpaired Mann-Whitney U tests. Data are from three independent experiments with $n=5$ per group. ** $p < 0.01$, *** $p < 0.001$, **** $p < 0.0001$.

19. Line 373-380 – Could you specify whether these analyses were done after Citrobacter challenge and if so when, or in mice that weren't challenged with Citrobacter at all? Its currently ambiguous from the positioning in the text and wording, so should be clarified

We appreciate the reviewer's insightful comment. The analyses were conducted in uninfected baseline animals, and we have clarified this in the respective sentence in line 375 of the clean version.

To investigate whether TXwildlings share this mechanistic ground, we applied the analytical framework established by Reese [47] and Beura [48] to transcriptomic data from uninfected lab mice, TXwildlings, and wildlings.

20. Line 378-380 – suggest make the specific pairwise comparisons clearer here e.g. “The adult human transcriptome was significantly enriched in the colon and lung transcriptome of TXwildlings and wildlings compared to lab mice, whereas lab mice showed a greater similarity to neonatal humans than TXwildlings or wildlings.”

Yes, your suggestion is appreciated and is a nice improvement to our manuscript (see line 379 of the clean version).

The adult human transcriptome was significantly enriched in the colon and lung transcriptome of TXwildlings and wildlings compared to lab mice, whereas lab mice showed a greater similarity to neonatal humans than TXwildlings or wildlings (Figure 6D).

21. Line 390 – Day 15 seems like peak weight loss, at least in SPF group

We thank the reviewer in highlighting this issue. We have changed this to show the weight on day 15 instead of day 14.

Point-by-point-response Figure 5

*Lab mice, TXwildlings, and wildlings were orally infected with *C. rodentium*, the body weight was analyzed at days 14 and 15 post-infection. Mean \pm SEM is shown. Data are from two independent experiments with $n=5$ mice per group. * $p<0.05$. Day 15 is now part of Figure 6a.*

22. Line 399 – Use “facility” instead of “location” to avoid confusion with location in GI tract
Thank you for this suggestion, we have changed the sentence accordingly in line 400 of the clean version.

However, their low resilience and location-facility-specific microbiota result in phenotypes that vary across research institutions.

23. Line 400-402 – Back up the claim of variability in phenotypes being caused by microbiome variability across facilities with references

We thank the reviewer for this suggestion and have added references to back up our claim (see line 403 of the clean version).

24. Line 408 – What is a “TXsystem” exactly?

We are thankful for raising this concern again. We addressed this previously in minor point #11.

25. Line 441 – Phenotype can be attributed to presence of, or absence of, the SFB?

We appreciate the reviewer pointing out this lack of clarity. All three groups are colonized with SFB, so the phenotype cannot be attributed to differences in the presence of SFB.

26. Line 454 – Reference [27] is maybe placed better with [51, 52] than at end of sentence

We thank the reviewer for raising this issue. We have grouped reference [27] (now [31]) with references [51, 52] (now [64, 65]) in line 456 of the clean version.

Since wildlings have been shown to phenocopy human responses and could have prevented failed clinical trials [31, 64, 65], it is plausible to hypothesize that TXwildlings may similarly enhance the safety and efficacy of translational research efforts, as demonstrated with wildlings [7].

27. Line 460 – What do key attributes I-V refer to? Refer back to intro

We appreciate this suggestion and have added a reference to the introduction (see line 461 of the clean version).

In this work, we presented an evidence-based rationale to use natural gut microbiota as candidate for standardization due to their key attributes mentioned in the introduction (I-V).

28. Line 463 – How would researchers get the microbiome for the TXwildling system, and how would they use it? Perhaps this statement is a bit too bold/premature for this stage of the research?

We appreciate the reviewer raising this critical issue. As mentioned in two of our responses above, it is our primary goal to foster the field of natural microbiota-based mouse models globally. Thus, glycerol stocks of natural microbiota are readily available from our biobank and we are happy to ship them out to laboratories as well as research institutions worldwide in order to facilitate the creation of TX wildlings in various facilities and on various genetic backgrounds. Further, as mentioned above, we have added a detailed paragraph in the “Material and methods” section that will ensure that other institutions can reproduce our model and successfully use it in various research contexts. Moreover, we have now rephrased the respective paragraph and emphasize that we present a proof-of-concept study. Please see line 459 of the clean version.

“Taken together, microbiota standardization addresses the reproducibility crisis by resolving the issue of divergent microbiota across institutions, though no suitable candidate has been identified to date [18]. In this work, we presented an evidence-based rationale and proof-of-concept study to use natural gut microbiota as candidate for standardization due to their key attributes mentioned in the introduction (I-V). We developed and validated the TXwildling system – a scalable, standardizable, and controlled approach to make natural microbiota-based models globally accessible on various genetic backgrounds. Although this need to be further validated and tested in different locations, we anticipate that the widespread adoption of TXwildlings will may enable the discovery of novel treatments, enhance reproducibility, reduce research costs, and improve the safety and success of translational efforts, ultimately advancing human health.”

29. Line 682 – Just a thought, but it would be interesting to analyze the post-infection composition of the TXwildling microbiome to see how it is changed by an infection challenge, especially in comparison to the SPF mice.

We agree with the reviewer that this would be an interesting analysis. However, exploring this topic would be beyond the scope of this manuscript. To provide insight, we have prepared a simple PCoA plot showing the microbiome of lab mice, TXwildlings and wildlings before and after C. rodentium infection. This plot shows that the microbiome of TXwildlings and wildlings is not greatly affected while that of lab mice may shift slightly.

Point-by-point-response Figure 6

Lab mice, TXwildlings, and wildlings were orally infected with *C. rodentium*. Fecal pellets were collected before and 28 days post-infection and subjected to 16S rRNA sequencing. Data are from two independent experiments with $n=5$ per group.

30. Line 700 – Typo “Golay” not “Golary” ?

We thank the reviewer for pointing out this typo, which we have corrected in line 554 of the clean version.

Briefly, DNA was normalized to 25 ng/ μ l and used for sequencing PCR with unique 12-base **Golary-Golay** barcodes incorporated via specific primers (obtained from Sigma).

31. Fig. 6D – found colours of points in key are a bit too transparent to differentiate.

We thank the reviewer for this thoughtful comment. We have changed the colors of the dots to be less transparent.

Point-by-point-response Figure 7

GSEA among the indicated mouse groups compared to a ranked gene list derived from healthy adult vs. neonatal PBMCs. Gene signatures include the top 500 significantly differently expressed genes of colon or lung tissues of TXwildlings vs. lab mice (dark blue), wildling vs. lab mice (green) or lab mice vs. TXwildlings (light blue). Data are from two independent experiments with $n=5$ per group. Graphs are shown in Figure 6d.

References Mentioned:

Chen, Y. H., Yeung, F., Lacey, K. A., Zaldana, K., Lin, J. D., Bee, G. C. W., ... & Cadwell, K. (2023). Rewilding of laboratory mice enhances granulopoiesis and immunity through intestinal fungal colonization. *Science immunology*, 8(84), eadd6910.

Kamada, Nobuhiko, et al. "Regulated virulence controls the ability of a pathogen to compete with the gut microbiota." *Science* 336.6086 (2012): 1325-1329.

Rosshart, S. P., Vassallo, B. G., Angeletti, D., Hutchinson, D. S., Morgan, A. P., Takeda, K., ... & Rehmann, B. (2017). Wild mouse gut microbiota promotes host fitness and improves disease resistance. *Cell*, 171(5), 1015-1028.

Reviewer #3 (Remarks to the Author):

1. Stappenbeck, T.S. and H.W. Virgin, *Accounting for reciprocal host-microbiome interactions in experimental science*. *Nature*, 2016. **534**(7606): p. 191-9.
2. Leung, J.M., et al., *Rapid environmental effects on gut nematode susceptibility in rewilded mice*. *PLoS Biol*, 2018. **16**(3): p. e2004108.
3. Beura, L.K., et al., *Normalizing the environment recapitulates adult human immune traits in laboratory mice*. *Nature*, 2016. **532**(7600): p. 512-6.
4. Ma, J., et al., *Laboratory mice with a wild microbiota generate strong allergic immune responses*. *Sci Immunol*, 2023. **8**(87): p. eadf7702.
5. Mansoori Moghadam, Z., et al., *Reactive oxygen species regulate early development of the intestinal macrophage-microbiome interface*. *Blood*, 2025.
6. Mann, J., et al., *The Microbiome Modifies Manifestations of Hemophagocytic Lymphohistiocytosis in Perforin-Deficient Mice*. *European Journal of Immunology*, 2025. **55**(1): p. e202451061.
7. Rosshart, S.P., et al., *Laboratory mice born to wild mice have natural microbiota and model human immune responses*. *Science*, 2019. **365**(6452).
8. Drude, N., et al., *A facility for laboratory mice with a natural microbiome at Charité – Universitätsmedizin Berlin*. *Lab Animal*, 2024. **53**(12): p. 351-354.
9. Rosshart, S.P., et al., *Wild Mouse Gut Microbiota Promotes Host Fitness and Improves Disease Resistance*. *Cell*, 2017. **171**(5): p. 1015-1028.e13.

10. Rehmann, B., et al., *Integrating natural commensals and pathogens into preclinical mouse models*. Nature Reviews Immunology, 2024.
11. Reese, T.A., et al., *Sequential Infection with Common Pathogens Promotes Human-like Immune Gene Expression and Altered Vaccine Response*. Cell Host Microbe, 2016. **19**(5): p. 713-9.
12. Underhill, D.M. and I.D. Iliev, *The mycobiota: interactions between commensal fungi and the host immune system*. Nature Reviews Immunology, 2014. **14**(6): p. 405-416.
13. Iliev, I.D. and I. Leonardi, *Fungal dysbiosis: immunity and interactions at mucosal barriers*. Nature Reviews Immunology, 2017. **17**(10): p. 635-646.
14. Virgin, H.W., *The virome in mammalian physiology and disease*. Cell, 2014. **157**(1): p. 142-50.
15. Norman, J.M., S.A. Handley, and H.W. Virgin, *Kingdom-Agnostic Metagenomics and the Importance of Complete Characterization of Enteric Microbial Communities*. Gastroenterology, 2014. **146**(6): p. 1459-1469.
16. Lim, E.S., D. Wang, and L.R. Holtz, *The Bacterial Microbiome and Virome Milestones of Infant Development*. Trends in Microbiology, 2016. **24**(10): p. 801-810.
17. Chudnovskiy, A., et al., *Host-Protozoan Interactions Protect from Mucosal Infections through Activation of the Inflammasome*. Cell, 2016. **167**(2): p. 444-456.e14.
18. Bruno, P., T. Schöler, and S.P. Rosshart, *Born to be wild: utilizing natural microbiota for reliable biomedical research*. Trends in Immunology.
19. Chung, H., et al., *Gut Immune Maturation Depends on Colonization with a Host-Specific Microbiota*. Cell, 2012. **149**(7): p. 1578-1593.
20. Seedorf, H., et al., *Bacteria from diverse habitats colonize and compete in the mouse gut*. Cell, 2014. **159**(2): p. 253-66.

„Laboratory mice engrafted with natural gut microbiota possess a wildling-like phenotype“

REVIEWERS' COMMENTS

Reviewer #1 (Remarks to the Author):

- The housing conditions seem to be described twice now: L490-496 and L545-550
 - Whilst L520 says "into only one out of 5 mice per cage", assuming the other mice were colonized via co-housing, L540/541 says that "each mouse was inoculated..."
 - L564: 16S sequencing. Always use the term "16S rRNA gene amplicon sequencing" or at least "16S rRNA amplicon sequencing".
 - "Shotgun metagenomic sequencing" is redundant. "Shotgun sequencing" is enough
- We thank the reviewer for carefully reading our manuscript and have changed the mentioned aspects.*

Reviewer #2 (Remarks to the Author):

The authors have responded comprehensively to all of our initial concerns, including conducting follow-up evaluations of pathogen screening and metagenomic analysis of mycobiome elements, etc., within their samples. Additionally, they have provided good justification for their decisions to use a pathogen-containing wildling source of their microbiome, and why they opted to construct a novel system as opposed to testing the existing WildR system. Finally, additional contextualization of methods and results (particularly the addition of statistics to many areas we thought they were missing) has been helpful in communicating the results more thoroughly.

We thank the reviewers for their valuable feedback throughout the revision process that helped to significantly improve our manuscript. We are pleased to hear that our additional analyses, methodological clarifications, and contextualizations have addressed the initial concerns. We appreciate the positive evaluation of our justifications and revisions, and are glad that the changes improved the clarity and completeness of our manuscript.

1. The rationale behind using wildlings and not WildR, and issues around pathogens have been comprehensively addressed, improving the paper's value and clarity. Our only remaining points on this are:

(i) it would be useful to clarify in the paper a little more clearly (as was done in responses) that a key 'hurdle' point for natural mouse microbiotas that the wildling model overcomes (compared to e.g. co-housing with pet store mice), is that it excludes HUMAN pathogens so can in theory be BSL1 and in normal mouse facilities. But that mouse pathogens intentionally remain, and you consider these an important component of the natural model.

[As a side note, one other hurdle likely to remain, is likely to be that many mouse facilities might still feel uneasy/not allow housing wildlings in the same building, as some of the mouse pathogens like Spironucleus are likely to be on whole-facility "Exclusion" lists, to avoid contamination risk to other precious or vulnerable mouse stocks.]

We thank the reviewer for pointing out this aspect. We have added an additional sentence in line 149 for clarifying this aspect.

"Of note, TXwildlings as well as wildlings do not harbor human pathogens and meet biosafety level 1 standards."

(ii) the metagenomic analyses of which non-bacterial microbes (fungi, viruses) were detected in the wildling and TXwildling don't seem to be fully presented, e.g. what other viruses and fungi were detected in the metagenomic screen that weren't targeted in the pathogen screening? This is currently only presented in summaries like "Fungi" or "Fungi-like protists" in Fig. 1e, but we can't see what's under the hood. Suggest include as this is important for anyone wishing to use the wildling microbiota in their studies. Perhaps a supplementary excel table tab?

We thank the reviewer for his detailed interest in our results. To provide full transparency on the non-bacterial components of the wildling and TXwildling microbiota, we have created Supplementary Table 1 with the top 20 taxa of each group.

2. "We have included statistical analyses in supplementary Figure S1".

Stats are better generally (thank you), tho a couple of points still:

- Please add to legend what statistical test was used in addition to the derived p-values.

We are thankful for pointing out this aspect and have added the statistical tests to all respective figure legends.

- To strengthen/show the point that lab mouse microbiomes are less variable after TX treatment (e.g. Fig. S1g), key stat/figure would be to compare the average pairwise inter-

facility (Taconic vs Jackson etc) microbiota distance and intra-facility distance before, and then after TX treatment. You would ideally show that (i) inter-facility before vs after comparison - the average inter-facility microbiome distance before TX treatment is much larger than after, and (ii) inter- vs. intra-facility distance comparison: the inter-facility distance far outstrips the intra-facility distance before TX treatment, but after TX treatment these are not statistically different (or far less so).

We are grateful for this insightful suggestion, which highlights a central finding of our study. To illustrate the reduction in inter-vendor variability after transplantation, we have added Figure S2b, which compares the distance between mice of different vendors before and after transplantation (see also Point-by-point-response Figure 1a). As illustrated, the inter-vendor distance decreased significantly following transplantation with wildling material. Comparing intra- versus inter-vendor comparison before and after treatment, we found that indeed they were very different before and very similar after transplantation (see Point-by-point response Figure 1b). We decided to focus on the more informative inter-vendor comparison to avoid overcomplicating the presentation.

Point-by-point-response Figure 1.

Lab mice were purchased from different vendors and transplanted with wildling fecal material. (a) Distance of mice of each vendor before and after transplantation to the centroid of the indicated vendor was calculated (inter-vendor variability). (b) Distance of mice of each vendor to the centroid of the indicated vendor were calculated before and after transplantation (inter-vendor variability) and compared to the distance to each vendors centroid (intra-vendor variability). Data are from one (Charles River, Envigo, Janvier, Jackson) or three (Taconic) independent experiments with five mice each. Statistical significances were calculated by unpaired, two-tailed Mann-Whitney U tests. Panel (a) is also part of Figure S2b.

3. While we agree that the biological relevance of alpha diversity is uncertain, we think including the additional figure on richness and compositional barplots of the different microbiotas that was included in response to reviewers would be useful for readers – can you include it as a supplementary figure?

We thank the reviewer for this excellent suggestion. We have split Figure S1 into two figures, with Figure S2 showing the suggested panels.

4. References: “This is evident in the divergent composition of their bacterial microbiome [30-35] and the underrepresentation of key non-bacterial constituents such as the virome and mycobiome [31,33, 36].” While the authors have added more refs for this point about natural microbiotas being different (30-35), these still seem to all or largely be using the wildling/WildR model in lab mice. What we meant is that there are quite a few papers that analyse other populations of gut microbiota in wild vs laboratory house mice, that could be cited here to broaden the point, since wildlings are from one site.

Examples would be:

- Wang, Jun, et al. "Dietary history contributes to enterotype-like clustering and functional metagenomic content in the intestinal microbiome of wild mice." *Proceedings of the National Academy of Sciences* 111.26 (2014): E2703-E2710.

- Kreisinger J, Cížková D, Vohánka J, Piálek J. Gastrointestinal microbiota of wild and inbred individuals of two house mouse subspecies assessed using high-throughput parallel pyrosequencing. *Mol Ecol*. 2014;23:5048–60.

- Bowerman KL, et al. Effects of laboratory domestication on the rodent gut Microbiome. *ISME Commun*. 2021;1:1–14.

- Hanski, Eveliina, et al. "Wild house mice have a more dynamic and aerotolerant gut microbiota than laboratory mice." *BMC microbiology* 25.1 (2025): 204.

We are thankful for this clarification. We have added the suggested references to the manuscript.

5. The figure in the responses showing how gavaged and co-housed mice vary in the speed with which they diverge from lab mouse microbiome over time is useful... it encourages 3Rs (refinement) as shows gavage is not necessarily needed to get a good colonisation, it just speeds it up a little. Suggest accommodate this in Fig. S1.

We agree with the reviewer and have added the suggested panel to Figure S1d.

6. Please clarify this in the legend of Fig. 6 also that the GSEA was performed on uninfected mice; with the infection challenge data the first panel in this figure, one could be forgiven for assuming the GSEA was also performed on the same mice.

We thank the reviewer for pointing out this potentially confusing aspect and have clarified things accordingly.

“GSEA among the indicated mouse **comparisons of uninfected lab mice, TXwildlings and wildlings** compared to a ranked gene list derived from healthy adult vs. neonatal PBMCs.”